# A causal view of compositional zero-shot recognition

**Yuval Atzmon**[1] **Felix Kreuk** [1,2] **Uri Shalit** [3] **Gal Chechik**[1,2]
[1]NVIDIA Research, Tel Aviv, Israel
[2]Bar-Ilan University, Ramat Gan, Israel
[3]Technion - Israel Institute of Technology
yatzmon@nvidia.com, gchechik@nvidia.com,

## Abstract

People easily recognize new visual categories that are new combinations of known components. This *compositional generalization* capacity is critical for learning in real-world domains like vision and language because the long tail of new combinations dominates the distribution. Unfortunately, learning systems struggle with compositional generalization because they often build on features that are *correlated* with class labels even if they are not "essential" for the class. This leads to consistent misclassification of samples from a new distribution, like new combinations of known components.

Here we describe an approach for compositional generalization that builds on causal ideas. First, we describe compositional zero-shot learning from a causal perspective, and propose to view zero-shot inference as finding "*which intervention caused the image?*". Second, we present a causal-inspired embedding model that learns disentangled representations of elementary components of visual objects from correlated (confounded) training data. We evaluate this approach on two datasets for predicting new combinations of attribute-object pairs: A well-controlled synthesized images dataset and a real-world dataset which consists of fine-grained types of shoes. We show improvements compared to strong baselines. Code and data are provided in `https://github.com/nv-research-israel/causal_comp`

## 1   Introduction

Compositional zero-shot recognition is the problem of learning to recognize new combinations of known components. People seamlessly recognize and generate new compositions from known elements and *Compositional Reasoning* is considered a hallmark of human intelligence [33, 34, 6, 4]. As a simple example, people can recognize a purple cauliflower even if they have never seen one, based on their familiarity with cauliflowers and with other purple objects (Figure 1b). Unfortunately, although *feature compositionality* is a key design consideration of deep networks, current deep models struggle when required to generalize to new *label compositions*. This limitation has grave implications for machine learning because the heavy tail of unfamiliar compositions dominates the distribution of labels in perception, language, and decision-making problems.

Models trained from data tend to fail with compositional generalization for two fundamental reasons: **distribution-shift** and **entanglement**. First, recognizing new combinations is an extreme case of distribution-shift inference, where label combinations at test time were never observed during training (zero-shot learning). As a result, models learn correlations during training that hurt inference at test time. For instance, if all cauliflowers in the training set are white, the correlation between the color and the class label is predictive and useful. A correlation-based model like (most) deep networks will learn to associate cauliflowers with the color white during training, and may fail when presented with a purple cauliflower at test time. For the current scope, we put aside the fundamental

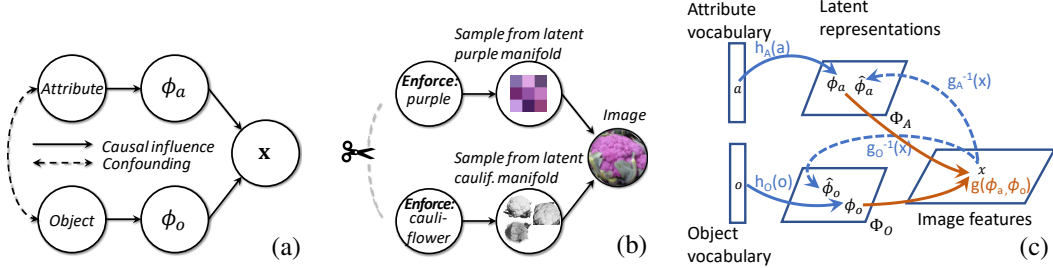

Figure 1: **(a)** The causal graph that generates an·image. The solid arrows represent the real-world processes by which the two categorical variables "Object" and "Attribute" each generate "core features" [21, 17] $\phi_o$ and $\phi_a$. The core features then jointly generate an image feature vector $\mathbf{x}$. The core features are assumed to be stable for unseen combinations of objects and attributes. The dotted double-edged arrows between the Object and Attribute nodes indicates that there is a process "confounding" the two: they are not independent of each other. **(b)** An intervention that generates a test image of a purple-cauliflower, by enforcing $a = purple$ and $o = cauliflower$. It cuts the confounding link between the two nodes [49] and changes the joint distribution of the nodes to the "*interventional distribution*". **(c)** Illustration of the learned mappings, detailed in Section 4

semantic question about what defines the class of an object (cauliflower), and assume that it is given or determined by human observers.

The second challenge is that the training samples themselves are often labeled in a compositional way, and disentangling their "elementary" components from examples is often an ill-defined problem [38]. For example, for an image labeled as white cauliflower, it is hard to tell which visual features capture being a cauliflower, and which, being white. In models that learn from data the representation of these terms may be inherently entangled, and it would be hard to separate which visual features represent white and which represent a cauliflower.

These two challenges are encountered when learning deep *discriminative* models from data. For example, consider a simple model that learns the concept "cauliflower", by training a deep model over all cauliflower images (VisProd [44]), and the same for the concept "white". At inference time, simply select the most likely attribute $\hat{a} = \arg\max_a p(a|\mathbf{x})$ and, independently, the most likely object $\hat{o} = \arg\max_o p(o|\mathbf{x})$. Unfortunately, this model, while quite powerful, tends to be sensitive to training-specific correlations in its input.

Here we propose to address compositional recognition by modelling images as being generated, or caused, by real-world entities (labels) (Figure 1). This model recognizes that the distribution $p(Image=\mathbf{x}|Attr=a, Obj=o)$ is more likely to be stable across the train and test environments $\left(p_{test}(\mathbf{x}|a, o) = p_{trn}(\mathbf{x}|a, o)\right)$ [54, 49, 57]: it means that unlike objects or attributes by themselves, *combinations* of objects and attributes generate the same distribution over images in train and test. We propose to consider images of unseen combinations as generated by interventions on the attribute and object labels. In causal inference, intervention means that the value of a random variable is forced to some value, without affecting its causes (but affecting other variables that depend on it, Figure 1b). We cast zero-shot inference as the problem of finding which intervention caused a given image.

In the general case, the conditional distribution $p(\mathbf{x}|a, o)$ can have arbitrary an complex structure and may be hard to learn. We explain how treating labels as causes, rather than as effects of the image, reveals an independence structure that makes it easier to learn $p(\mathbf{x}|a, o)$. We propose conditional independence constraints applied to the structure of this distribution and show how the model can be learned effectively from data.

The paper makes the following novel contributions: First, we provide a new formulation of compositional zero-shot recognition using a causal perspective. Specifically, we formalize inference as a problem of finding the most likely intervention. Second, we describe a new embedding-based architecture that infers causally stable representations for compositional recognition. Finally, we demonstrate empirically that in two challenging datasets, our architecture better recognizes new unseen attribute-object compositions compared to previous methods.

## 2 Related work

**Attribute - object compositionality:** [42] studied decomposing attribute-object combinations. They embedded attributes and object classes using deep networks pre-trained on other large datasets. [45] proposed to view attributes as linear operators in the embedding space of object word-embeddings. Operators are trained to keep transformed objects similar to the corresponding image representation. [46] proposed a method similar to [42] with an additional decoding loss. [60] used GANs to generate a feature vector from the label embedding. [50] trained a set of network modules, jointly with a gating network that rewires the modules according to embeddings of attribute and object labels. [37] is a very recent framework inspired by group theory that incorporates symmetries in label-space.

**Compositional generalizations:** Several papers devised datasets to directly evaluate compositional generalization for vision problems by creating a test set with new combinations of train-set components. [25] introduced a synthetic dataset inherently built with compositionality splits. [1, 9] introduced new compositional splits of VQA datasets [2] and show that the performance of existing models degrades under their new setting. [26] used a knowledge graph for composing classifiers for verb-noun pairs. [6] proposed an experimental framework for measuring compositional generalization and showed that structured prediction models outperform image-captioning models.

**Zero-shot Learning (ZSL):** Compositional generalization can be viewed as a special case of zero-shot learning [62, 35, 8], where a classifier is trained to recognize (new) unseen classes based on their semantic description, which can include a natural-language textual description [48] or predefined attributes [35, 7]. To discourage attributes that belong to different groups from sharing low-level features, [24] proposed a group-sparse regularization term.

**Causal inference for domain adaptation:** Several recent papers take a causal approach to describe and address domain adaptation problems. This includes early work by [55] and [64]. Adapting these ideas to computer vision, [17] were one of the first papers to propose a causal DAG describing the generative process of an image as being generated by a "domain", which generates a label and an image. They use this graph for learning invariant components that transfer across domains. [36, 3] extended [17] with adversarial training [15]. It learned a single discriminative classifier, $p(o|\mathbf{x})$, that is robust against domain shifts and accounts for the dependency of classes on domains. When viewing attributes as "domains", one of the independence terms in their model corresponds to one term (c) in Eq. (6). [28, 55] discusses image recognition as an "anti-causal" problem, inferring causes from effects. [10, 27] studied learning causal structures under sparse distributional shifts. [40] proposed to learn causal relationships in images by detecting the causal direction of two random variables. [14] used targeted interventions to address distribution shifts in imitation learning. [32] learned a conditional-GAN model jointly with a causal-model of label distribution. [5] proposed a regularization term to improve robustness against distributional changes. Their view is complementary to this paper in that they model the labeling processes, where images cause labels, while our work focuses on the data generating process (labels cause images). [21] proposed a similar causal DAG for images, while adding *auxiliary* information such as that some images are of the *same instance* with a different "style". This allowed the model to identify core features. The approach described in this paper does not use such auxiliary information. It also views the "object" and "attribute" as playing mutual roles, making their inferred representations invariant to each-other.

**Unsupervised disentanglement of representations:** Several works use a VAE [31] approach for unsupervised disentanglement of representations [39, 22, 11, 13, 52, 41]. This paper focuses on a different problem setup: (1) Our goal is to infer a joint attribute-object pair, disentangling the representation is a useful byproduct. (2) In our setup, attribute-object combinations are dependent in the training data, and new combinations may be observed at test time. (3) We do not use unsupervised learning. (4) We take a simpler embedding based approach.

## 3 Method overview

We start with a descriptive overview of our approach. For simplicity, we skip here the causal motivation and describe the model in an informal way from an embedding viewpoint.

Our model is designed to estimate $p(\mathbf{x}|a, o)$, the likelihood of an image feature vector $\mathbf{x}$, conditioned on a tuple $(a, o)$ of attribute-object labels. For inference, we iterate over all combinations of labels and select $(\hat{a}, \hat{o}) = \mathrm{argmax}_{a,o}\, p(\mathbf{x}|a, o)$.

**To estimate** the distribution $p(\mathbf{x}|a, o)$, our model learns two embedding spaces: $\Phi_A$ for attributes, and $\Phi_O$ for objects (see Figure 1a). These spaces can be thought of as semantic embedding spaces, where an attribute $a$ (say, "white") has some dense prototypical representation $\phi_a \in \Phi_A$, and an object $o$ (say, a cauliflower) has a dense representation $\phi_o \in \Phi_o$. Given a new image $\mathbf{x}$, we learn a mapping to three spaces. First, an inferred attribute embedding $\hat{\phi}_a \in \Phi_A$ represents the attribute seen in the image (say, how white is the object). Second, an inferred object embedding $\hat{\phi}_o \in \Phi_O$ represents the object seen in the image (say, how "cauliflowered" it is). Finally, we also represent the image in a general space of image features.

**Learning** the parameters of the model involves learning the three mappings above. In addition, we learn the representation of the attribute prototype ("White") $\phi_a \in \Phi_A$ and the representation of the object prototype ("Cauliflower") $\phi_o \in \Phi_O$. Very naturally, we want that a perceived attribute $\hat{\phi}_a$ would be embedded close to its attribute prototype $\phi_a$. Our loss captures this intuition. Finally, we also aim to have the representation spaces of attributes and objects statistically independent. The intuition is that we want to keep the representation of an object (cauliflower) independent of the attribute (white), so we can recognize that object when seen with new attributes (purple cauliflower).

At this point, the challenge remains to build a principled model that can be learned efficiently from data. We now turn to the formal and detailed description of the approach.

## 4   A causal formulation of compositional zero-shot recognition

We put forward a causal perspective that treats labels as causes of an image, rather than its effects. This direction of dependencies is consistent with the mechanism underlying natural image generation and, as we show below, allows us to recognize unseen label combinations.

Figure 1 presents our causal generative model. We consider two "elementary factors" which are categorical variables called "Attribute" $a \in \mathcal{A}$ and "Object" $o \in \mathcal{O}$, and are dependent (confounded) in the training data. As one example, European swans (Cygnus) are white but Australian ones are black. The data collection process may make 'white' and 'swan' confounded if collected in Europe, even-though this dependency does not apply in Australia.. The model also has two semantic representation spaces: one for attributes $\Phi_A = \mathbb{R}^{d_A}$ and another for objects $\Phi_O = \mathbb{R}^{d_O}$. An attribute $a$ induces a distribution $p(\phi_a|a)$ over the representation space, which we model as a Gaussian distribution. We denote by $h_a$ a function that maps a categorical attributes to the center of this distribution in the semantic space $h_a : \mathcal{A} \to \Phi_A$ (Figure 1c). The conditional distribution is therefore $\phi_a \sim \mathcal{N}(h_a, \sigma_a^2 I)$. We have a similar setup for $p(\phi_o|o) \sim \mathcal{N}(h_o, \sigma_o^2 I)$.

Given the semantic embedding of the attribute and object, the probability of an image feature vector $\mathbf{x} \in \mathcal{X}$ is determined by the representations $p(\mathbf{x}|\phi_a, \phi_o)$, which we model as Gaussian, w.r.t a mapping $g$, $\mathbf{x} \sim \mathcal{N}(g(\phi_a, \phi_o), \sigma_x^2 I)$. $\phi_a$ and $\phi_o$ can be viewed as an encoding of "core features", namely encoding a representation of attribute and object that is "stable" in the training set and test set, as proposed by [21, 17]. Namely, the conditional distributions $p(\phi_a|a)$ and $p(\phi_o|o)$ do not substantially change for unseen combinations of attributes and objects.

We emphasize that our causal graph is premised on the belief that what we use as objects and attributes are truly distinct aspects of the world, giving rise to different core features. For attributes that have no physical meaning in the world, it may not be possible to postulate two distinct processes giving rise to separate core features.

### 4.1   Interventions on elementary factors

Causal inference provides a formal mechanism to address the confounding effect through a "do-intervention"[1]. A "Do-intervention" *overrides* the joint distribution $p_{trn}(a, o)$, enforcing $a, o$ to specific values and propagates them through the causal graph. With this propagation, an intervention *changes the joint distribution* of nodes in the graph. Therefore, a test image is generated according to a new joint-distribution, denoted in causal language as the *interventional distribution* $p^{do(A=a,O=o)}(\mathbf{x})$. Thus, for zero-shot learning, we postulate that inference about a test image is equivalent to asking: **Which intervention on attributes and objects caused the image?**

# 5 Inference

We propose to infer the attribute and object by choosing the most likely interventional distribution:

$$(\hat{a}, \hat{o}) \quad = \quad \underset{a,o \in \mathcal{A} \times \mathcal{O}}{\operatorname{argmax}} \; p^{do(A=a,O=o)}(\mathbf{x}). \tag{1}$$

This inference procedure is more stable than the discriminative zero-shot approach, since the generative conditional distribution is equivalent to the interventional distribution [55].

$$p^{do(A=a,O=o)}(\mathbf{x}) = p(\mathbf{x}|a,o). \tag{2}$$

This holds both for training and test, so we simply write $p(\mathbf{x}|a,o)$. This likelihood depends on the core features $\phi_A, \phi_O$ which are latent variables; Computing the likelihood exactly requires to marginalize (integrate) over the latent variables. Since this integral is very hard to compute, we take a "hard" approach instead, evaluating the integrand at its most likely value. Since $\phi_A, \phi_O$ are not known, we estimate them from the image $x$, by learning a mapping function $\hat{\phi}_a = g_A^{-1}(x)$ (see Figure 1c). The supplemental describes these approximation steps in details. It shows that the negative log-likelihood $-\log p(\mathbf{x}|a,o)$ can be approximated by

$$\hat{L}(a,o) = \frac{1}{\sigma_a^2}||\hat{\phi}_a - h_a||^2 + \frac{1}{\sigma_o^2}||\hat{\phi}_o - h_o||^2 + \frac{1}{\sigma_x^2}||\mathbf{x} - g(h_a, h_o)||^2 \quad . \tag{3}$$

Here, $h_a, h_o$ and $g(h_a, h_o)$ are the parameters of the Gaussian distributions of $\phi_a, \phi_o$ and $\mathbf{x}$. The factors $a$ and $o$ are inferred by taking the $\operatorname{argmin}_{a,o} \hat{L}(a,o)$ of Eq. (3). Note that in the causal graph, $\phi_a, \phi_o$ are parent nodes of the image $\mathbf{x}$, but $\hat{\phi}_a, \hat{\phi}_o$ are estimated from $x$ and are therefore child nodes of $\mathbf{x}$, and therefore do not immediately follow the conditional independence relations that $\phi_a, \phi_o$ obey. We elaborate on this point in section 6.

# 6 Learning

Our model consists of five learned mappings: $h_A, h_O, g, g_A^{-1}$ and $g_O^{-1}$, illustrated in Figure 1c. All mappings are modelled using MLPs. We aim to learn the parameters of these mappings such that the (approximated) negative log-likelihood of Eq. (3) is minimized. In addition, we also include in the objective several regularization terms designed to encourage properties that we want to induce on these mappings. Specifically, the model is trained with a linear combination of three losses.

$$\mathcal{L} = \mathcal{L}_{data} + \lambda_{indep}\mathcal{L}_{indep} + \lambda_{invert}\mathcal{L}_{invert}, \tag{4}$$

where $\lambda_{indep} \geq 0$ and $\lambda_{invert} \geq 0$ are hyperparameters. We now discuss these losses in detail.

**(1) Data Likelihood loss.** The first component of the loss, $\mathcal{L}_{data}$, corresponds to the (approximate) negative log likelihood of the model, as described by Eq. (3)

$$\mathcal{L}_{data} = ||h_a - g_A^{-1}(\mathbf{x})||^2 + ||h_o - g_O^{-1}(\mathbf{x})||^2 + \lambda_{ao}\mathcal{L}_{triplet}\Big(\mathbf{x}, (a,o), (a,o)_{neg}\Big). \tag{5}$$

For easier comparisons with [44], we replaced the rightmost term in Eq. (3) with the standard triplet loss $\mathcal{L}_{triplet}$ with Euclidean distance $||\mathbf{x} - g(h_a, h_o)||^2$. $\lambda_{ao} \geq 0$ is a hyperparameter.

**(2) Independence loss.** The second component of the loss $\mathcal{L}_{indep}$ is designed to capture conditional-independence relations and apply them to the reconstructed core factors $\hat{\phi}_a, \hat{\phi}_o$. By that, the following property is encouraged: $p^{do(O=o)}(\hat{\phi}_o) \approx p^{do(A=a,O=o)}(\hat{\phi}_o)$ and $p^{do(A=a)}(\hat{\phi}_a)) \approx p^{do(A=a,O=o)}(\hat{\phi}_a)$. Namely, learning a representation of objects that is robust to attribute interventions, and vice versa.

In more detail, the causal graph (Figure 1a) dictates conditional-independence relations for the latent core factors $\phi_a, \phi_o$:

$$\begin{array}{llll} (a) & \phi_a \perp\!\!\!\perp O | A = a & (b) & \phi_a \perp\!\!\!\perp \phi_o | A = a, \\ (c) & \phi_o \perp\!\!\!\perp A | O = o & (d) & \phi_a \perp\!\!\!\perp \phi_o | O = o. \end{array} \tag{6}$$

These relations reflect the independent mechanisms that generate the training data.

Since the core factors $\phi_a, \phi_o$, are latent and not observed, we wish that their reconstructions $\hat{\phi}_a$ and $\hat{\phi}_o$ maintain approximately the same independence relations. For example, we encourage $(\hat{\phi}_o \perp\!\!\!\perp A | O = o)$ to capture the independence in Eq. (6)c.

To learn mappings that adhere to these statistical independences over $\hat{\phi}_a$ and $\hat{\phi}_o$, we regularize the learned mappings using a differentiable measure of statistical dependence. Specifically, we use the Hilbert-Schmidt Information Criterion (HSIC) [19, 20]. HSIC is a non-parametric method for estimating the statistical dependence between samples of two random variables, based on an implicit embedding into a universal reproducing kernel Hilbert space. In the infinite-sample limit, the HSIC between two random variables is 0 if and only if they are independent [20]. HSIC also has a simple finite-sample estimator which is easily calculated and is differentiable w.r.t. the input variables. In supplemental Section B, we describe the details of $\mathcal{L}_{indep}$ and how it is optimized with HSIC, and why minimizing $L_{indep}$ indeed encourages the property $p^{do(O=o)}(\hat{\phi}_o) \approx p^{do(A=a,O=o)}(\hat{\phi}_o)$. This minimization can be viewed as minimizing the "Post Interventional Disagreement" (PIDA) metric of [58], a recently proposed measure of disentanglement of representations. We explain this perspective in more detail in the supplemental (B.2).

There exist alternative measures for encouraging statistical independence, such as adversarial training [15, 36, 3] or techniques based on mutual information [16, 53, 29]. HSIC has the benefit that it is non-parametric and therefore does not require training an additional network. It was easy to optimize, and was provide useful in previous literature [59, 56, 43, 18].

**(3) Invertible embedding loss.** The third component of the loss, $\mathcal{L}_{invert}$, encourages the label-embedding mappings $h_a$, $h_o$, and $g(h_a, h_o)$ to preserve information about their source labels when minimizing $\mathcal{L}_{data}$. Without this term, minimizing $||\hat{\phi}_a - h_a||^2$ may reach trivial solutions because the model has no access to ground-truth values for $\phi_a$ or $h_a$ (same for $\phi_o, h_o$). Similar to [44], we use a cross-entropy (CE) loss with a linear layer that classifies the labels that generate each embedding, and a hyperparameter $\lambda_g$:

$$\mathcal{L}_{invert} = CE(a, f_a(h_a)) + CE(o, f_o(h_o)) + \lambda_g \big[ CE(a, f_{ga}(g(h_a, h_o))) + CE(a, f_{go}(g(h_a, h_o))) \big].$$

## 7 Experiments

### 7.1 Data

Despite several studies of compositionality, current datasets used for evaluations are quite limited. Two main benchmarks were used in evaluations of previous literature: *MIT states* [23] and *UT-Zappos50K* [63].

The MIT-states dataset was labeled automatically using early technology of image search engine based on text surrounding images. As a result, labels are often incorrect. We quantified label quality using human raters , and found that they are too noisy to be useful for proper evaluations. In more detail, we presented images to human raters, along with candidate attribute labels from the dataset. Raters were asked to select the best and second-best attributes that describe the noun (multiple-choice setup). Only 32% of raters selected the correct attribute for their first choice (top-1 accuracy), and only 47% of raters had the correct attribute in one of their choices (top-2 accuracy). The top-2 accuracy was only slightly higher than adding a random label on top of the top-1 label (yielding 42%). To verify that raters were attentive, we also injected 30 "sanity" questions that had two "easy" attributes, yielding top-2 accuracy of 100%. See supplemental (H) for further details. We conclude that this level of $\sim 70\%$ label noise is too noisy for evaluating noun-attribute compositionality.

**Zappos:** We evaluate our approach on the Zappos dataset, which consists of fine-grained types of shoes, like "leather sandal" or "rubber sneaker". It has 33K images, 16 attribute classes, and 12 object classes. We use the split of [50] and the provided ResNet18 pretrained features. The split contains both seen pairs and unseen pairs for validation and test. It uses 23K images for training of 83 seen pairs, a validation set with 3K images from 15 seen and 15 unseen pairs, and a test set with 3K images from 18 seen and 18 unseen pairs. All the metrics we report for our approach and compared baselines are averaged over 5 random initializations of the model.

**AO-CLEVr:** To evaluate compositional methods on a well-controlled clean dataset, we generated a synthetic-images dataset containing images of "easy" Attribute-Object categories. We used the CLEVr framework [25], hence we name the dataset *AO-CLEVr*. AO-CLEVr has attribute-object pairs created from 8 attributes: { red, purple, yellow, blue, green, cyan, gray, brown } and 3 objects {sphere, cube, cylinder}, yielding 24 attribute-object pairs. Each pair consists of 7500 images. Each image has a single object that consists of the attribute-object pair. The object is randomly assigned

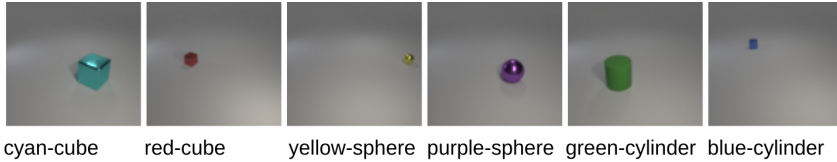

Figure 2: Example images of AO-CLEVr dataset and their labels.

cyan-cube    red-cube    yellow-sphere  purple-sphere  green-cylinder  blue-cylinder

one of two sizes (small/large), one of two materials (rubber/metallic), a random position, and random lightning according to CLEVr defaults. See Figure 2 for examples.

For cross-validation, we used two types of splits. The first uses the same unseen pairs for validation and test. This split allows us to quantify the potential generalization capability of each method. In the second split, which is harder, unseen validation pairs are not overlapping with the unseen test pairs. Importantly, we vary the ratio of unseen:seen pairs on a range of (2:8, 3:7, …,7:3), and for each ratio we draw 3 random seen-unseen splits. We report the average and the standard error of the mean (S.E.M.) over the three random splits and three random model initialization for each split. We provide more details about the splits in the suppl.

## 7.2 Compared methods

**(1) Causal**. Our approach as described in Section 5. For Zappos it also *learns* a single layer network to project the pretrained image features to the feature space $\mathcal{X}$. We also evaluate a variant named **Causal** $\lambda_{indep}$=0, which nulls the loss terms that encourage the conditional independence relations

**(2) VisProd**: A common discriminatively-trained baseline [45, 42]. It uses two classifiers over image features to predict the attribute and object independently, and approximates $p(a, o|\mathbf{x}){\sim}p(a|\mathbf{x})p(o|\mathbf{x})$.

**(3) VisProd&CI**: A discriminatively-trained variant of our model. We use VisProd as a vanilla model, regularized by the conditional independence loss $\mathcal{L}_{indep}$, where we use the top network layer activations of attributes and objects as proxies for $\hat{\phi}_a, \hat{\phi}_o$.

**(4) LE:** *Label embedding* [45] trains a neural network to embed images and attribute-object labels to a joint feature space. LE is a vanilla baseline because it approximately models $p(\mathbf{x}|a, o)$, but without modelling the core-features.

**(5) ATTOP:** *Attributes-as-operators* [45] views attributes as operators over the embedding space of object label-embeddings. We use the code of [45] to train ATTOP and LE.

**(6) TMN:** *Task-modular-networks* [50] trains a set of network modules jointly with a gating network. The gate rewires the modules according to embeddings of attributes and objects. We used the implementation provided by the authors and followed their grid for hyperparameter search (details in suppl.). Our results differ on Zappos because we report an average over 5 random initializations rather than a single initialization as reported in [50]. Some random initializations reproduce well their reported AUC metric.

We explicitly avoid using prior knowledge in the form of *pretrained* label embeddings, because we are interested to quantify the effectiveness of our approach to naturally avoid unreliable correlations. Yet, most of the methods we compare with, rely on pretrained embeddings. Thus, we provide additional results using random initialization for the compared methods, denoted by an asterisk (e.g. LE*).

**Implementation details** of our approach and reproduced baselines are given in the supplemental.

## 7.3 Evaluation

In zero-shot (ZS) attribute-object recognition, a training set $\mathcal{D}$ has $N$ labeled samples of images: $\mathcal{D} = \{(\mathbf{x}_i, a_i, o_i), i = 1 \ldots N\}$, where each $\mathbf{x}_i$ is a feature vector, $a_i \in \mathcal{A}$ is an attribute label, $o_i \in \mathcal{O}$ is an object label and each pair $(a_i, o_i)$ is from a set of (source) *seen* pairs $\mathcal{S} \subset \mathcal{A} \times \mathcal{O}$. At test time, a new set of samples $\mathcal{D}' = \{\mathbf{x}_i, i = N + 1 \ldots N + M\}$ is given from a set of target pairs. The target set is a union of the set of seen pairs $\mathcal{S}$ with new *unseen* pairs $\mathcal{U} \subset \mathcal{A} \times \mathcal{O}, \mathcal{U} \cap \mathcal{S} = \emptyset$. Our goal is to predict the correct pair of each sample.

**Evaluation metrics:** We evaluate methods by the accuracy of their top-1 prediction for recognizing seen and unseen attribute-object pairs. In AO-CLEVr, we compute the balanced accuracy across pairs, namely, the average of per-class accuracy. This is the common metric to evaluate zero-shot

benchmarks [61, 62]. Yet, in Zappos, we used the standard (imbalanced) accuracy, to be consistent with the protocol of [50],

We compute metrics in two main evaluations setups, which differ in their *output* label space, namely, which classes can be predicted. *(1) Closed*: Predictions can be from unseen class-pairs *only*. *(2) Open*: Predictions can be from all pairs in the dataset, seen or unseen. This setup is also known as the *generalized* zero-shot learning setup [62, 12]. Specifically, we compute: **Seen**: Accuracy is computed on *samples from* seen class-pairs. **Unseen**: Accuracy is computed on *samples from* unseen class-pairs. **Harmonic mean**: A metric that quantifies the overall performance of both Open-Seen and Open-Unseen accuracy. It is defined as: $H = 2(Acc_{seen} * Acc_{unseen})/(Acc_{seen} + Acc_{unseen})$. For the harmonic metric, we follow the protocol of [61, 62], which does not take an additional post-processing step. We note that some papers [50, 37, 12] used a different protocol averaging between seen and unseen. Finally, we report the **Area Under Seen-Unseen Curve (AUSUC)**, which uses a post-processing step [12] to balance the seen-unseen accuracy. Inspired by the area-under-the-curve procedure, it adjusts the level of confidence of seen pairs by adding (or subtracting) a constant (see [12] for further details). To compute AUSUC, we sweep over a range of constants and compute the area under the seen-unseen curve.

For early stopping, we use (i) The Harmonic for the open setup. (ii) The closed accuracy for the closed setup. In Zappos, we followed [50] and use the AUSUC for early stopping at the closed setup.

All experiments were performed on a cluster of DGX-V100 machines. Training a single model for 1000 epochs on the $5 : 5$ AO-CLEVr split (with ~80K samples) takes 2-3 hours.

## 8 Results

We describe here the results obtained on AO-CLEVr (overlapping-split) and Zappos. Additional results are reported in the supplemental, including the full set of metrics and numeric values; using random initialization; results with the non-overlapping split (showing a similar trend to the overlapping split); studying our approach in greater depth through ablation experiments; and an error analysis.

**AO-CLEVr:** Figure 3 (right) shows the Harmonic metric for AO-CLEVr for the whole range of unseen:seen ratios. Unsurprisingly, the more seen pairs are available, the better all models perform for unseen combinations of attributes and objects. Our approach "Causal", performs better than or equivalent to all the compared methods. VisProd easily confuses the Unseen classes. ATTOP, is better than LE on the open unseen pairs but performs substantially worse than all methods on the seen pairs. TMN performs equally well as our approach for splits with mostly seen pairs but degrades when the fraction of seen pairs is below 4:6.

**The seen-unseen plane:** By definition, our task aims to perform well in two different metrics (multi-objective): accuracy on seen pairs and unseen pairs. It is therefore natural to compare approaches by their performance on the seen-unseen plane. This is important, because different approaches may select different operating points to trade-off accuracy on unseen for accuracy on seen. Figure 3 (left) shows how the compared approaches trade-off the metrics for the 5:5 split. ATTOP tends to favor unseen-pairs accuracy over the accuracy of seen pairs, vanilla models like VisProd, LE tend to favor seen classes. **Importantly**, it reveals that modelling the core-features largely improves the unseen accuracy, without hurting much the seen accuracy. Specifically, comparing *Causal* to vanilla baseline *LE*, improves the unseen acc. from 26% to 47% and reduces the seen acc. from 86% to 84%. Comparing *VisPros&CI* to *VisProd* improves the unseen acc. from 19% to 38% and reduces the seen Acc. from 85% to 82%.

| | UNSEEN | SEEN | HARMONIC | CLOSED | AUSUC |
|---|---|---|---|---|---|
| WITH PRIOR EMBEDDINGS | | | | | |
| LE | $10.7 \pm 0.8$ | $52.9 \pm 1.3$ | $17.8 \pm 1.1$ | $55.1 \pm 2.3$ | $19.4 \pm 0.3$ |
| ATTOP | $22.6 \pm 2.9$ | $35.2 \pm 2.7$ | $26.5 \pm 1.4$ | $52.2 \pm 1.8$ | $20.3 \pm 1.8$ |
| TMN | $9.7 \pm 0.6$ | $51.9 \pm 2.4$ | $16.4 \pm 1.0$ | $\mathbf{60.9 \pm 1.1}$ | $\mathbf{24.6 \pm 0.8}$ |
| NO PRIOR EMBEDDINGS | | | | | |
| LE* | $15.6 \pm 0.6$ | $52.0 \pm 1.0$ | $24.0 \pm 0.7$ | $58.1 \pm 1.2$ | $22.0 \pm 0.9$ |
| ATTOP* | $16.5 \pm 1.5$ | $15.8 \pm 1.9$ | $15.8 \pm 1.4$ | $42.3 \pm 1.5$ | $16.7 \pm 1.1$ |
| TMN* | $6.3 \pm 1.4$ | $\mathbf{55.3 \pm 1.6}$ | $11.1 \pm 2.3$ | $58.4 \pm 1.5$ | $24.5 \pm 0.8$ |
| CAUSAL $\lambda_{indep}$=0 | $22.5 \pm 2.0$ | $45.5 \pm 3.7$ | $29.4 \pm 1.5$ | $55.3 \pm 1.1$ | $22.2 \pm 0.9$ |
| CAUSAL | $\mathbf{26.6 \pm 1.6}$ | $39.7 \pm 2.2$ | $\mathbf{31.8 \pm 1.7}$ | $55.4 \pm 0.8$ | $23.3 \pm 0.3$ |

Table 1: Results for Zappos. $\pm$ denotes the Standard Error of the Mean (S.E.M.) over 5 random model initializations.

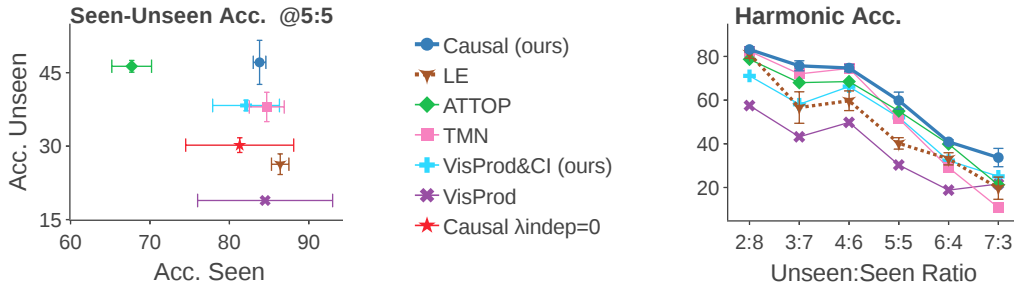

Figure 3: **Left:** The seen-unseen plane for the 5:5 split. Modelling the core features largely improves the unseen accuracy: Compare *Causal* to *LE* or to $\lambda_{indep}$=0 and compare *VisPros&CI* to *VisProd*. Error bars denote Standard Error of the Mean (S.E.M.) over 3 random splits and three random seeds. **Right:** Harmonic mean of seen-unseen accuracy for AO-CLEVR on a range of 20% to 70% unseen classes. To reduce visual clutter, error bars are shown only for our *Causal* method and for a vanilla baseline (*LE*).

**Zappos:** Our approach improves the Unseen and Harmonic metrics. For the "Closed" and "AUSUC" metrics it loses compared to TMN. We note that results on the Closed metric are less interesting from a causal point of view: A model cannot easily rely on the knowledge of which attribute-object pairs tend to appear in the test set.

For both AO-CLEVr and Zappos, the independence loss improves recognition on unseen pairs but hurts recognition of seen pairs. This is a known and important trade-off when learning models that are robust for interventions [51]. The independence loss discourages certain types of correlations, hence models *do not* benefit from them when the test and train distributions are identical (seen-pairs). However, the loss is constructed in such a way that these are exactly the correlations that fail to hold once the test distribution changes (unseen-pairs). Thus, ignoring these correlations improves performance for samples of unseen-pairs.

## 9   Discussion

We present a new causal perspective on the problem of recognizing new attribute-object combinations in images. We propose to view inference in this setup as answering the question "which intervention on attribute and object caused the image". This perspective gives rise to a causal-inspired embedding model. The model learns disentangled representations of attributes and objects although they are dependent in the training data. It provides better accuracy on two benchmark datasets.

The trade-off between seen accuracy and unseen accuracy reflects the fact that prior information about co-occurrence of attributes and objects in the training set is useful and predictive. A related problem has been studied in the setting of (non-compositional) generalized zero-shot learning [8]. We suggest that some of these techniques could be adapted to the compositional setting.

Several aspects of the model can be further extended by relaxing its assumptions. First, the assumption that image features are normally distributed may be limiting, and alternative ways to model this distribution may improve the model accuracy. Second, the model is premised on the prior knowledge that the attributes and objects have distinct and stable generation processes. However, this prior knowledge may not always be available, or some attribute may not have an obvious physical meaning. E.g. "cute", "comfortable" or "sporty", and in a multi-label setting [47] it is challenging to reveal what are the independent generation mechanisms themselves from confounded training data.

This paper focused on the case where attributes and objects are fully disentangled. Clearly, in natural language, many attributes and object are used in a way are that makes them dependent. For example, white wine is actually yellow, and the attribute a small bear is larger than a large bird. It remains an interesting question to extend the fully disentangled case to learn specific dependencies while leveraging the power of disentangled representations.

## Broader Impact

Compositional generalization, the key challenge of this work, is critical for learning in real-world domains where the long tail of new combinations dominates the distribution, like in vision-and-language tasks or for the perception modules of autonomous driving.

A causality-based approach, like the one we propose, may allow vision systems to make more robust inference, and debias correlations that naturally exist in the training data, allowing to use vision systems in complex environments where the distribution of labels and their combinations is varying. It has been shown in the past that vision systems may emphasize biases in the data, and the ideas put forward in the paper may help make systems more robust to such biases.

Such approach may be useful for improving fairness in various applications, for example by providing a more balanced visual recognition of individuals from minority groups.

## Acknowledgements

We thank Daniel Greenfeld, Idan Schwartz, Eli Meirom, Haggai Maron, Lior Bracha and Ohad Amosy for their helpful feedback on the early version. Uri Shalit was partially supported by the Israel Science Foundation (grant No. 1950/19).

## Funding Disclosure

Uri Shalit was partially supported by the Israel Science Foundation. Yuval Atzmon was supported by the Israel Science Foundation and Bar-Ilan University during his Ph.D. studies.

## Footnotes

[1]Our formalism and model can be extended to include other types of intervention on the joint distribution of attributes and objects. For simplicity, we focus here on the most-common "do-intervention".

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
