[Supplementary Material]

# Supplementary Information: A causal view of compositional zero-shot recognition

## A    Approximating $\mathrm{argmax}_{a,o}\, p(\mathbf{x}|a,o)$

The conditional likelihood $p(\mathbf{x}|a,o)$ can be written by marginalizing over the latent factors $\phi_a$ and $\phi_o$

$$p(\mathbf{x}|a,o) \;=\; \iint_{\phi_a,\phi_o} p(\mathbf{x},\phi_a,\phi_o|a,o) = \iint_{\phi_a,\phi_o} p(\mathbf{x}|\phi_a,\phi_o)p(\phi_a|a)p(\phi_o|o)d\phi_o d\phi_a\,. \quad (\text{S.7})$$

Computing this integral is hard in the general case. Variational autoencoders (VAEs) [31] approximate a similar integral by learning a smaller support using an auxiliary probability function $Q$ over the latent space. Here we make another approximation, taking a "hard" approach and find the single most likely integrand.

$$\underset{(a,o)\in\mathcal{A}\times\mathcal{O}}{\mathrm{argmax}} \; p(\mathbf{x}|\phi_a,\phi_o)p(\phi_a|a)p(\phi_o|o) \qquad (\text{S.8})$$

Based on Section 4, the three factors of the distributions are Gaussians. Therefore, maximizing Eq. (S.8) is equivalent to minimizing the negative log likelihood

$$\underset{(a,o)\in\mathcal{A}\times\mathcal{O}}{\mathrm{argmin}} \; \frac{1}{\sigma_x^2}||\mathbf{x} - g(\phi_a,\phi_o)||^2 + \frac{1}{\sigma_a^2}||\phi_a - h_a||^2 + \frac{1}{\sigma_o^2}||\phi_o - h_o||^2\,. \qquad (\text{S.9})$$

This expression is composed of three components. The components allow to infer $(a,o)$ by evaluating distances in three representation spaces $\mathcal{X}$, $\Phi_A$ and $\Phi_O$ (Section 4).

However, we cannot apply Eq. (S.9) to infer $(a,o)$ because the core features $\phi_a, \phi_o$ are latent. Next, we introduce two additional approximations we use to apply Eq. (S.9). The first, approximates $||\mathbf{x} - g(\phi_a,\phi_o)||^2$ by $||\mathbf{x} - g(h_a,h_o)||^2$, using a Taylor expansion at the means $(\phi_a,\phi_o) = (h_a,h_o)$. The second, recovers (infers) the core features $\phi_a, \phi_o$ from the image, and substitute the recovered features in $||\phi_a - h_a||^2$, $||\phi_o - h_o||^2$.

### A.1    Approximating $||\mathbf{x} - g(\phi_a,\phi_o)||^2$

A causal model can be equivalently represented using a "Structural Causal Model" (SCM) [49]. An SCM matches a set of assignments to a causal graph. Each node in the graph is assigned a deterministic function of its parent nodes and an independent noise term. Specifically, based on the Gaussian assumptions in Section 4, the SCM of our causal graph (Figure 1a) is

$$\phi_a \;=\; n_a + h_a \qquad (\text{S.10})$$
$$\phi_o \;=\; n_o + h_o \qquad (\text{S.11})$$
$$\mathbf{x} \;=\; n_x + g(\phi_a,\phi_o),\, , \qquad (\text{S.12})$$

where $n_a, n_o, n_x$ are jointly independent Gaussian random variables $n_a \sim \mathcal{N}(0,\sigma_a^2 I)$, $n_o \sim \mathcal{N}(0,\sigma_o^2 I)$, $n_x \sim \mathcal{N}(0,\sigma_x^2 I)$. $n_a, n_o, n_x$ represent sampling from the manifold of attributes, objects and images near their prototypes $h_a$ and $h_o$.

We use a zeroth-order Taylor expansion of $g(\phi_a,\phi_o)$ at $(\phi_a,\phi_o) = (h_a,h_o)$ and make the following approximation

$$\big(\mathbf{x} - g(\phi_a,\phi_o)\big) \approx \big(\mathbf{x} - g(h_a,h_o)\big) \quad . \qquad (\text{S.13})$$

Next, we discuss the first-order *approximation error*. For brevity, we denote with $\phi_{ao}$ the concatenation of the elements in $(\phi_a,\phi_o)$ into a single vector. Similarly $(h_a,h_o)$ into $h_{ao}$ and $(n_a,n_o)$ into $n_{ao}$.

We approximate $g(\phi_{ao})$ by a first-order Taylor expansion at $\phi_{ao} = h_{ao}$ to

$$g(\phi_{ao}) \approx g(h_{ao}) + [Jg](h_{ao}) \cdot (\phi_{ao} - h_{ao}) = g(h_{ao}) + [Jg](h_{ao}) \cdot n_{ao}, \qquad (\text{S.14})$$

where $[Jg]$ is the Jacobian of $g$ and the last equality stems from the SCM (Eqs. S.10, S.11).

Using Eq. (S.14) and the Cauchy-Schwarz inequality, the first-order squared error approximation of Eq. (S.13) is:

$$\mathbb{E}||\big(\mathbf{x} - g(\phi_{ao})\big) - \big(\mathbf{x} - g(h_{ao})\big)||^2 = \mathbb{E}||g(\phi_{ao}) - g(h_{ao})||^2$$
$$\approx \mathbb{E}||[Jg](h_{ao})n_{ao}||^2 \leq ||[Jg](h_{ao})||_F^2 \mathbb{E}||n_{ao}||^2 \tag{S.15}$$

This implies that the error of the approximation Eq. (S.13) is mainly dominated by the gradients of $g$ at $h_{ao}$, and the variance of $n_{ao}$. If the gradients and the variance of $n_{ao}$ are too large, then this approximation may be too coarse, and one may resort to more complex models like variational methods [31]. Empirically, we observe that this approximation is useful.

### A.2 Recovering the core features

To apply Eq. (S.9), we approximate $\phi_a, \phi_o$, by reconstructing (inferring) them from the image.

The main assumption we make is that images preserve the information about the core features they were generated from, at least to a level that allows us to differentiate what were the semantic prototypes ($h_a$ and $h_o$) of the core feature. Otherwise, inference on test data and labeling the training data by human raters will render to random guessing.

Therefore, we assume that there exists an inverse mappings that can infer $\phi_a$ and $\phi_o$ from the image up to a reasonably small error:

$$\hat{\phi}_a \equiv g_A^{-1}(\mathbf{x}) = \phi_a + \epsilon_A(x) \tag{S.16}$$
$$\hat{\phi}_o \equiv g_O^{-1}(\mathbf{x}) = \phi_o + \epsilon_O(x), \tag{S.17}$$

where $\epsilon_A(x), \epsilon_O(x)$ denote the image-based error of inferring the attribute and object core features.

Specifically, we assume that the error in substituting $\phi_a$ by $\hat{\phi}_a$ and $\phi_o$ by $\hat{\phi}_o$ in Eq. (S.9) is small enough to keep them close to their prototypes ($h_a$ and $h_o$). Namely, we make the following approximation

$$||\phi_a - h_a||^2 = ||\hat{\phi}_a - \epsilon_A(x) - h_a||^2 \leq ||\hat{\phi}_a - h_a||^2 + ||\epsilon_A(x)||^2 \approx ||\hat{\phi}_a - h_a||^2$$
$$||\phi_o - h_o||^2 = ||\hat{\phi}_o - \epsilon_O(x) - h_o||^2 \leq ||\hat{\phi}_o - h_o||^2 + ||\epsilon_O(x)||^2 \approx ||\hat{\phi}_o - h_o||^2. \tag{S.18}$$

To conclude, we use Eq. (S.13) and Eq. (S.18) to approximate Eq. (S.9) by :

$$\underset{(a,o)\in\mathcal{A}\times\mathcal{O}}{\mathrm{argmin}} \ \frac{1}{\sigma_x^2}||\mathbf{x} - g(h_a, h_o)||^2 + \frac{1}{\sigma_a^2}||\hat{\phi}_a - h_a||^2 + \frac{1}{\sigma_o^2}||\hat{\phi}_o - h_o||^2,$$

which is the expression for Eq. (3) in the main paper.

## B  Independence Loss

Our loss includes a component $\mathcal{L}_{indep}$, which is designed to capture the conditional-independence relations that the causal graph dictates (Eq. (6)). We now describe $\mathcal{L}_{indep}$ in detail.

Since we do not have the actual values of the latent core features $\phi_a$, $\phi_o$, we wish that their reconstructions $\hat{\phi}_a$ and $\hat{\phi}_o$ maintain approximately the same independence relations as Eq. (6). To learn mappings that adhere to these statistical independences over $\hat{\phi}_a$ and $\hat{\phi}_o$, we regularize the learned mappings using a differentiable measure of statistical dependence.

Specifically, we use a positive differentiable measure of the statistical dependence, denoted by $\mathcal{I}$. For two variables $(u, v)$, conditioned on a categorical variable $Y$, we denote by $\mathcal{I}(u, v|Y)$ the positive differentiable measure of the statistical conditional dependence of $(u, v|Y)$. For example we encourage approaching the equality in Eq. 6b by minimizing $\mathcal{I}(\hat{\phi}_a, \hat{\phi}_o|A)$. $\mathcal{I}$ is based on measuring the Hilbert-Schmidt Information Criterion (HSIC) [19, 20], which is a non-parametric method for estimating the statistical dependence between samples of two variables. We adapt the HSIC criterion to measure *conditional* dependencies. $\mathcal{I}$ penalizes conditional dependencies in a batch of samples $B = \big\{(u_i, v_i, y_i)\big\}_{i=1}^{|B|}$, by summing over groups of samples that have the same label:

$$\mathcal{I}(u,v|Y) = \frac{1}{|Y|}\sum_{y\in Y}\text{HSIC}(U|Y=y, V|Y=y) \tag{S.19}$$

$$\text{where} \quad (U|Y=y, V|Y=y) = \big\{(u_i,v_i)\in B \mid y_i = y\big\}.$$

Finally, we have $\mathcal{L}_{indep}$, a loss term that encourages the four conditional independence relations of 6:

$$\mathcal{L}_{indep} = \mathcal{L}_{oh} + \lambda_{rep}\mathcal{L}_{rep} \tag{S.20}$$

$$\mathcal{L}_{oh} = \mathcal{I}\big(\hat{\phi}_a, O|A\big) + \mathcal{I}\big(\hat{\phi}_o, A|O\big) \tag{S.21}$$

$$\mathcal{L}_{rep} = \mathcal{I}\big(\hat{\phi}_a, \hat{\phi}_o|A\big) + \mathcal{I}\big(\hat{\phi}_a, \hat{\phi}_o|O\big), \tag{S.22}$$

where $\lambda_{rep}$ is a hyper parameter.

Minimizing Eq. (S.21) encourages the representation of the inferred attribute $\hat{\phi}_a$, to be invariant to the categorical ("one-hot") representation of an object $O$. Minimizing Eq. (S.22) encourages $\hat{\phi}_a$ to be invariant to the "appearance" of an object ($\hat{\phi}_o$).

## B.1 An expression for Hilbert-Schmidt Information Criterion (HSIC) with linear kernel

The linear-kernel HSIC between two batches of vectors $\mathbf{U}, \mathbf{V}$ is calculated in the following manner [20]: It uses two linear kernel matrices $K_{i,j} = \mathbf{u}_{[i,:]}\mathbf{u}_{[j,:]}^T$, $L_{i,j} = \mathbf{v}_{[i,:]}\mathbf{v}_{[j,:]}^T$. Then the HSIC is calculated as the scaled Hilbert-Schmidt norm of their cross-covariance matrix:

$$HSIC(U,V) = \frac{1}{(n-1)^2}\cdot\mathbf{tr}(KHLH),$$

where $H_{ij} = \delta_{i,j} - \frac{1}{n}$ is a centering matrix , and $n$ is the batch size

## B.2 Minimizing $\mathcal{L}_{indep}$ encourages robustness of the reconstructed core features $\hat{\phi}_a, \hat{\phi}_o$.

The conditional-independence term of $\mathcal{L}_{oh}$ within $\mathcal{L}_{indep}$ is related to a metric named "Post Interventional Disagreement" (PIDA), recently introduced by [58]. PIDA measures disentanglement of representations for models that are trained from unsupervised data. This section explains their relation in more detail, by showing that minimizing $\mathcal{L}_{indep}$ encourages the following properties: $p^{do(O=o)}(\hat{\phi}_o)\approx p^{do(A=a,O=o)}(\hat{\phi}_o)$ and $p^{do(A=a)}(\hat{\phi}_a)\approx p^{do(A=a,O=o)}(\hat{\phi}_a)$.

The PIDA metric for attributes is measured by

$$PIDA(a'|a,o) := d\big(\mathbb{E}^{do(a)}[\hat{\phi}_{a'}], \mathbb{E}^{do(a,o)}[\hat{\phi}_{a'}]\big), \tag{S.23}$$

where $d$ is loosely described as "a suitable" positive distance function (like the $L_2$ distance). $PIDA(a'|a,o)$ quantifies the shifts in the inferred features $\hat{\phi}_{a'}$ when the object is enforced to $o$. Similarly, the PIDA term for objects is $PIDA(o'|o,a)$.

Below we show that encouraging the conditional independence ($\hat{\phi}_a \perp\!\!\!\perp O|A$), is equivalent to minimizing $d\big(p^{do(a)}(\hat{\phi}_{a'}), p^{do(a,o)}(\hat{\phi}_{a'})\big)$ and therefore minimizes $PIDA(a'|a,o)$.

First, in our causal graph (Figure 1a) a do-intervention on both $a$ and $o$ is equivalent to conditioning on $(a,o)$

$$p^{do(a,o)}(\hat{\phi}_{a'}) = p(\hat{\phi}_{a'}|a,o). \tag{S.24}$$

Second, minimizing Eq. (S.21) encourages the conditional independence ($\hat{\phi}_{a'} \perp\!\!\!\perp O|A$), which is equivalent to minimization of $d\big(p(\hat{\phi}_{a'}|a), p(\hat{\phi}_{a'}|a,o)\big)$ [2]. Third, when $d\big(p(\hat{\phi}_a|a), p(\hat{\phi}_a|a,o)\big)$ approaches zero, then $p^{do(a)}(\hat{\phi}_a)$ approaches $p(\hat{\phi}_a|a)$. This stems from the adjustment formula, $p^{do(a)}(\hat{\phi}_{a'}) = \sum_o p(\hat{\phi}_{a'}|a,o)p(o)$: When $d\big(p(\hat{\phi}_{a'}|a), p(\hat{\phi}_{a'}|a,o)\big)$ approaches zero, then

$\sum_o p(\hat{\phi}_{a'}|a,o)p(o)$ approaches $\sum_o p(\hat{\phi}_{a'}|a)p(o) = p(\hat{\phi}_{a'}|a)$. Therefore $p^{do(a)}(\hat{\phi}_{a'})$ approaches $p(\hat{\phi}_{a'}|a)$.

As a result, we have the following: Minimizing Eq. (S.21) leads to $p^{do(a,o)}(\hat{\phi}_{a'})$ approaching $p(\hat{\phi}_{a'}|a)$, which as we have just shown, leads to $p(\hat{\phi}_{a'}|a)$ approaching $p^{do(a)}(\hat{\phi}_{a'})$. Therefore, $d\big(p^{do(a)}(\hat{\phi}_{a'}), p^{do(a,o)}(\hat{\phi}_{a'})\big)$ is minimized and accordingly, $PIDA(a'|a,o)$ (Eq. S.23) is minimized.

Similarly, for objects, encouraging the conditional independence $(\hat{\phi}_{o'} \perp\!\!\!\perp A|O)$ minimizes $PIDA(o'|a,o)$. Therefore, minimizing $\mathcal{L}_{oh}$ (Eq. S.21), optimizes both $PIDA(a'|a,o)$ and $PIDA(o'|a,o)$.

## C   Implementation details

### C.1   Architecture

Similar to LE [44], we implemented $g$, $h_a$ and $h_o$ by MLPs with ReLU activation. For $g_A^{-1}$ and $g_O^{-1}$, every layer used a batch-norm and leaky-relu activation. All the MLPs share the same size of hidden layer, denoted by $d_h$.

For experiments on Zappos, we also learned a single layer network $f$ to project pretrained image features to the feature space $\mathcal{X}$. This strategy was inspired by the baseline models LE and ATTOP. Learning a projection $f$ finds better solutions on the validation set than using the pretrained features as $\mathcal{X}$.

For HSIC we used the implementation of [18] and applied a linear kernel as it does not require tuning additional hyper-parameters.

### C.2   Optimization

**AO-CLEVr**: We optimized AO-CLEVr in an alternating fashion: First we trained $h_a$, $g_A^{-1}$ and $g$, keeping $h_o$, $g_O^{-1}$ frozen. Then we froze $h_a$, $g_A^{-1}$ and trained $h_o$, $g_O^{-1}$ and $g$. This optimization strategy allows to stabilize the attribute representation when minimizing Eq. (S.22). The strategy was developed during the early experiments with a low-dimensional ($\mathcal{X} \subset \mathbb{R}^2$) synthetic dataset. In the ablation study (Section F below), we show that a standard (non-alternating) optimization strategy achieves comparable results, but with somewhat higher bias toward seen accuracy.

We used SGD with Nesterov Momentum to train with AO-CLEVr. Empirically, we found that SGD allowed finer control over $\mathcal{L}_{indep}$ than Adam [30].

In practice, we weighed the loss of $||\hat{\phi}_a - h_a||^2$ and $||\hat{\phi}_o - h_o||^2$ according to the respective attribute and object frequencies in the training set. This detail has a relatively small effect on performance. Without weighing the loss, the Harmonic decreases by 1.2% (from 68.8% to 67.6%) and Unseen accuracy decreases by 1.3% (from 57.5% to 56.2%).

**Zappos**: We optimized Zappos in a standard (non-alternating) fashion. We couldn't use the *alternating* optimization strategy, because in Zappos we also learn a mapping $f$ that projects pretrained image features to the feature space $\mathcal{X}$. Thus, updating the parameters of $f$ changes the mapping to $\mathcal{X}$ and we cannot keep $h_a$, $g_A^{-1}$ frozen once $\mathcal{X}$ changes.

As in [44], we used Adam [30] to train with Zappos.

### C.3   Early Stopping and Hyper-parameter selection

We trained each model instantiation for a maximum of 1000 epochs and early stopped on the validation set.

We used two metrics for early-stopping and hyper-parameter selection on the validation set: (i) the Harmonic metric for testing the unseen-accuracy, the seen-accuracy and the Harmonic; and (ii) accuracy of the Closed metric for testing the Closed accuracy. In Zappos, we followed [50] and used the AUSUC for testing both the AUSUC metric and the Closed accuracy.

For our approach and all the compared methods, we tuned the hyper-params by first taking a coarse random search, and then further searching around the best performing values on the validation set. As a rule of thumb, we first stabilized the hyper-parameters of the learning-rate, weight-decay, and architecture. Then we searched in finer detail over the hyper-parameters of the loss functions. At the most fine-grained iteration of the random search, each combination of hyper-parameters was evaluated with 3 different random weight initializations, and metrics were averaged over 3 runs. We chose the set of hyper-parameters that maximized the average metric of interest.

Since AO-CLEVr has $6 \cdot 3 = 18$ different splits, we searched the hyper-parameters over a single split of each of the ratios {2:8, 5:5, 6:4, 7:3}. For {3:7, 4:6} ratios, we used the hyper-parameters chosen for the {5:5} split.

## C.4   Hyper-parameters for loss function

In practice, to weigh the terms of the loss function, we use an equivalent but different formulation. Specifically, we set $\lambda_{indep}=1$ and $\lambda_{invert}=1$, and use the following expressions for $\mathcal{L}_{indep}$ and $\mathcal{L}_{invert}$

$$\mathcal{L}_{indep} = \lambda_{oh}\mathcal{L}_{oh} + \lambda_{rep}\mathcal{L}_{rep} \tag{S.25}$$

$$\mathcal{L}_{invert} = \lambda_{icore}\big(CE(a, f_a(h_a)) + CE(o, f_o(h_o))\big) + \tag{S.26}$$
$$\lambda_{ig}\big(CE(a, f_{ga}(g(h_a, h_o))) + CE(a, f_{go}(g(h_a, h_o)))\big).$$

where $\lambda_{oh}, \lambda_{rep}, \lambda_{icore}$ and $\lambda_{ig}$ weigh the respective loss terms.

## C.5   Grid-search ranges

For *Causal* with AO-CLEVr, we started the random grid-search over the following ranges: **Architecture:** (1) Number of layers for $h_a$ and $h_o \in \{0, 1, 2\}$, (2) Number of layers for $g_A^{-1}$ and $g_O^{-1} \in \{1, 2, 3\}$, (3) Number of layers for $g \in \{2, 4\}$, (4) Common size of hidden layers $d_h \in \{10, 30, 150, 300, 1000\}^3$. **Optimization:** (1) learning rate $\in \{$1e-5, 3e-5, 1e-4, 3e-4, 1e-3$\}$, when using alternate training, we used different learning rates for each alternation. (2) weight-decay $\in \{$1e-5, 1e-4, 1e-3, 0.01, 0.1, 1$\}$ (3) $\lambda_{rep} \in \{0, 0.001, 0.003, 0.01, 0.03, 0.1, 0.3, 1, 3, 10, 30, 100, 300\}$ (4) $\lambda_{oh} \in \{0, 0.001, 0.003, 0.01, 0.03, 0.1, 0.3, 1, 3, 10, 30, 100, 300\}$ (5) $\lambda_{ao} \in \{0, 0.1, 0.3, 1, 3, 10\}$ (6) $\lambda_{icore} \in \{0, 0.01, 0.03, 0.1, 0.3, 1, 3, 10, 30, 100\}$ (7) $\lambda_{ig} \in \{0, 0.01, 0.03, 0.1, 0.3, 1, 3, 10, 30, 100\}$. We didn't tune batch-size, we set it to 2048.

For *Causal* with Zappos, we used some hyper-parameters found with Causal&AO-CLEVr and some already known for LE with Zappos. Specifically, for the architecture, we set (1) Number of hidden layers for $h_a$ and $h_o = 0$ (linear embedding), (2) Number of layers for $g_A^{-1}$ and $g_O^{-1} = 2$, (3) Number of layers for $g = 2$ (4) Common size of hidden layers $d_h = 300$. For the optimization, in order to find a solution around the solution used by LE, we selected $\lambda_{ao} = 1000, \lambda_{ig} = 0$, weight-decay $= $ 5e-5, batch-size $= 2048$; and $\lambda_{icore} = 100$ as with Causal&AO-CLEVr. We applied the random search protocol over $\lambda_{rep} \in 15 \cdot \{0, 0.001, 0.003, 0.01, 0.03, 0.1, 0.3, 1, 3, 10, 30, \}, \lambda_{oh} \in 15 \cdot \{0, 0.001, 0.003, 0.01, 0.03, 0.1, 0.3, 1, 3, 10, 30, \}$, and learning rate $\in \{$1e-4, 3e-4$\}$

For *TMN*, we applied the random grid-search according to the ranges defined in the supplemental of [50]. Specifically: lr $\in \{0.0001, 0.001, 0.01, 0.1\}$, lrg $\in \{0.0001, 0.001, 0.01, 0.1\}$, batch-size $\in \{64, 128, 256, 512\}$, concept-dropout $\in \{0, 0.05, 0.1, 0.2\}$, nmod $\in \{12, 18, 24, 30\}$, output-dimension $\in \{8, 16\}$, number-of-layers $\in \{1, 2, 3, 5\}$. Additionally, we trained TMN for a maximum of 30 epochs, which is $\times 6$ longer than the recommended length (4-5 epochs). As instructed, we chose the number of negatives to be "all negatives". With Zappos, our grid-search found a hyper-parameters' combination with better performance than the one reported by the authors.

For *ATTOP* with AO-CLEVr, we used the following ranges: $\lambda_{aux} \in \{0, 0.001, 0.003, 0.01, 0.03, 0.1, 0.3, 1, 3, 10, 30, 100\}$, $\lambda_{comm} \in \{0, 0.001, 0.003, 0.01, 0.03, 0.1, 0.3, 1, 3, 10, 30, 100\}$,

emb-dim $\in \{10, 30, 150, 300, 1000\}$, weight-decay $\in \{$1e-5, 1e-4, 1e-3, 1e-2$\}$. We used the default learning rate 1e-4, and used a batch size of 2048.

For *ATTOP* with Zappos, we used the hyper-parameters combinations recommended by [44, 50], and also searched for emb-dim $\in \{300, 1000\}$

For *LE*, we used the following ranges: weight-decay $\in \{$1e-4, 1e-3, 1e-2$\}$, emb-dim $\in \{10, 30, 150, 300, 1000\}$. With Zappos we also followed the guideline of [50] and searched lr $\in \{0.0001, 0.001\}$.

Note that for ATTOP and LE, when Glove embedding is enabled then emb-dim is fixed to 300.

For *VisProd* with AO-CLEVr, we used the following ranges: We used the same number of layers, weight-decay and learning rates as used to train "Causal". We also followed the alternate-training protocol. With Zappos, we searched for emb-dim $\in \{100, 300, 1000\}$, lr $\in \{$1e-4, 1e-3$\}$.

### C.6 AO-CLEVr dataset

**Pretrained features:** Similar to Zappos, we extracted pretrained features for AO-CLEVr using a pretrained ResNet18 CNN.

**Cross validation splits:** For cross-validation, we used two types of splits. The first uses the same unseen pairs for validation and test. We call this split the "overlapping" split. The split allows us to quantify the potential generalization capability of each method. The second split, is harder, where unseen validation pairs are not overlapping with the unseen test pairs. We call this split the "*non*-overlapping" split.

For the overlapping split, we varied the ratio of unseen:seen pairs on a range of (2:8, 3:7, . . . 7:3), and for each ratio we drew 3 random seen-unseen splits. For the non-overlapping split, we varied the ratio of unseen:seen pairs on a range of (2:6, 3:5, . . . 5:3), and for each ratio we drew 3 random seen-unseen splits. In addition, we always draw 20% of the pairs for validation.

## D  AO-CLEVr with a non-overlapping validation set.

Here we present results for AO-CLEVr with the "*non*-overlapping" split. For this split, the unseen validation pairs are not overlapping with the unseen test pairs. It is harder than the "overlapping" split, which uses the same unseen pairs for validation and test.

The non-overlapping split is important because, in practice, we cannot rely on having labeled samples of the unseen pairs for validation purposes.

Figure S.1 shows the measured metrics when comparing "Causal", LE, TMN, and ATTOP and varying the ratio of seen:unseen pairs between 2:6 to 5:3.

For the main zero-shot metrics (Open-Unseen, Harmonic and Closed), our approach "Causal", performs substantially better than the compared methods. ATTOP performs substantially worse on "seen" pairs.

Figure S.1: AO clevr with a *non*-overlaping validation set.

## E  Complete results for AO clevr with overlapping split

Figure S.2 shows the accuracy metrics for compared approaches with AO-CLEVR when varying the fraction of seen:unseen classes (between 2:8 to 7:3). The top row in the figure shows the measured

metrics. The bottom row shows the difference (subtraction) from LE. We selected LE as the main reference baseline because its embedding loss approximately models $p(\mathbf{x}|a, o)$, but without modeling the core-features.

Figure S.2: Accuracy metrics for AO-CLEVR on a sweep of 20% unseen classes up to 70% unseen classes. The top row show the measured metrics. The bottom row show the difference (*subtraction*) of measured metrics from the LE baseline method. Error bars denote Standard Error of the Mean (S.E.M) over 3 random splits. To reduce visual clutter, error bars are shown only for our Causal method and for the reference baseline (LE).

Across the full sweep of unseen:seen ratios, our approach "*Causal*", performs better than or equivalent to all the compared methods for the main zero-shot metrics (Open-Unseen, Harmonic and Closed). VisProd, which approximates $p(a, o|\mathbf{x})$, has a relatively low Unseen accuracy. *VisProd&CI*, the discriminatively-trained variant of our model, improves the Unseen performance by a large margin, while not hurting VisProd Seen accuracy. ATTOP is better than LE on open unseen pairs but performs substantially worse than all methods on the seen pairs. TMN performs equally well as our approach for splits with mostly seen pairs (unseen:seen @ 2:8, 3:7, 4:6), but degrades when the fraction of seen pairs reduces below 60%.

### E.1 Baseline models without language embeddings

Figure S.3 compares LE, ATTOP, and TMN with and without initialization by Glove word embedding. It demonstrates that for AO-CLEVr, Glove initialization somewhat hurts LE, improves ATTOP, and is mostly equivalent to TMN.

Figure S.3: Comparing LE, ATTOP, and TMN with and without initialization by Glove word embedding. Glove initialization somewhat hurts LE, improves ATTOP, and is mostly equivalent for TMN.

## F   Ablation study

To understand the contribution of the different components of our approach, we conducted an ablation study to quantify the effect of the components. We report test metrics for one of the 5:5 "overlapping" splits of AO-CLEVr. Specifically, the split used for hyper-parameter search.

| | UNSEEN | SEEN | HARMONIC | CLOSED |
|---|---|---|---|---|
| CAUSAL | **57.5 ± 2.3** | 85.7 ± 3.4 | **68.8 ± 2.1** | 73.8 ± 1.0 |
| $\lambda_{indep} = 0$ | 30.0 ± 1.5 | **97.2 ± 0.2** | 45.7 ± 1.8 | 68.4 ± 1.0 |
| $\lambda_{ao} = 0$ | 44.9 ± 1.9 | 83.5 ± 5.6 | 58.1 ± 0.3 | 72.9 ± 3.0 |
| $\lambda_{invert} = 0$ | 19.5 ± 2.2 | 46.5 ± 2.9 | 27.3 ± 2.3 | 28.5 ± 4.6 |
| NON-ALTERNATE TRAINING | | | | |
| CAUSAL | 53.3 ± 2.0 | 90.6 ± 2.3 | 67.0 ± 1.5 | **74.5 ± 1.2** |
| $\lambda_{rep} = 0$ | 53.0 ± 4.7 | 91.3 ± 1.4 | 66.7 ± 3.3 | 70.7 ± 1.3 |
| $\lambda_{oh} = 0$ | 52.8 ± 4.0 | 90.5 ± 0.6 | 66.5 ± 3.0 | 71.9 ± 2.0 |
| $\lambda_{indep} = 0$ | 38.0 ± 2.3 | 94.6 ± 0.4 | 54.1 ± 2.4 | 68.9 ± 0.6 |

| | UNSEEN | SEEN | HARMONIC | CLOSED |
|---|---|---|---|---|
| WITH PRIOR EMBEDDINGS | | | | |
| LE | 21.4 ± 1.1 | 84.1 ± 1.8 | 34.0 ± 1.3 | 34.2 ± 2.4 |
| ATTOP | 48.7 ± 0.5 | 73.5 ± 0.8 | 58.5 ± 0.1 | 58.2 ± 0.5 |
| TMN | 32.3 ± 2.8 | 87.3 ± 4.1 | 47.0 ± 3.3 | 65.1 ± 3.0 |
| NO PRIOR EMBEDDINGS | | | | |
| VISPROD | 19.1 ± 1.3 | 94.3 ± 1.1 | 31.7 ± 1.8 | 60.1 ± 0.2 |
| LE* | 28.2 ± 1.7 | 87.5 ± 0.5 | 42.5 ± 1.9 | 43.4 ± 2.7 |
| ATTOP* | 45.6 ± 0.5 | 76.3 ± 1.0 | 57.0 ± 0.1 | 54.6 ± 1.3 |
| TMN* | 36.6 ± 5.2 | 89.1 ± 3.5 | 51.6 ± 5.6 | 66.5 ± 4.3 |
| VISP&CI | 40.5 ± 2.7 | 84.7 ± 4.3 | 54.4 ± 1.8 | 59.9 ± 0.3 |

Table S.1: **Left:** Ablation study on a 5:5 split of AO-CLEVr. We use the split used for hyper-param search. ± denote S.E.M on 3 random initializations. **Right:** Reference metrics for baselines.

Table S.1 reports the test metrics when ablating different components of our approach: We first compared the different components of the model while using alternate-training (see implementation details). Next, we compared the alternate-training strategy to standard (non-alternate) training. Finally, we compared the different components of the conditional-independence loss.

We compared the following components:

1. "Causal" is our approach described in Section 4. We tested is with both alternate training, and standard (non-alternate) training.

2. $\lambda_{indep} = 0$ indicates nullifying the loss terms that encourage the conditional independence relations.

3. $\lambda_{ao} = 0$ indicates nullifying the embedding to the image space $\mathcal{X}$.

4. $\lambda_{invert} = 0$ indicates nullifying the term that preserves information about source labels of the attribute and object embeddings.

5. $\lambda_{rep} = 0$ indicates nullifying $\mathcal{L}_{rep}$: The term that encourages invariance between $\hat{\phi}_a$ to $\hat{\phi}_o$.

6. $\lambda_{oh} = 0$ indicates nullifying $\mathcal{L}_{oh}$: The term that encourages invariance of $\hat{\phi}_a$ to a categorical representation of an object (and similarly for $\hat{\phi}_o$).

First, we find that both alternate-training and standard non-alternate training result in a comparable Harmonic metric. However, alternate training has a better Unseen accuracy (57.5% vs 53.3%), but lower Seen accuracy (85.7% vs 90.6%)

Second, nullifying each of the major components of the loss has a substantial impact on the performance of the model. Specifically, (1) nullifying $\lambda_{indep}$ reduces the Harmonic from 68.8% to 45.7%, (2) nullifying $\lambda_{ao}$ reduces the Harmonic to 58.1%, (3) nullifying $\lambda_{invert}$ reduces the Harmonic to 27.3%.

Finally, $\mathcal{L}_{oh}$ and $\mathcal{L}_{rep}$ have a synergistic effect on the performance of AO-CLEVr. Their individual performance metrics are comparable, but jointly they improve the Closed accuracy from ~71.5% to 74.5%.

**Accuracy versus hidden layer size:** Figure S.4 shows the different accuracy metrics when changing the hidden layer size for the 5:5 split of AO-CLEVr. It shows that the seen accuracy increases with the layer size, presumably because it can capture more subtleties of the seen pairs. The unseen accuracy appears to be bi-modal, with peaks at 15-units and 150 units. Indeed the cross-validation procedure selected a layer size of 150, because it maximized the harmonic-mean of the seen and unseen accuracy.

# G   Error analysis

## G.1   Zappos

We analyzed the *errors* that *Causal* makes when recognizing unseen pairs in Zappos (open). In 53% of cases the object was predicted correctly, in 14% the attribute was predicted correctly and in

Figure S.4: Ablation: Accuracy versus hidden layer size (in a logarithmic scale)

33% neither. It appears that in Zappos, recognizing the object transferred better to the unseen pairs, presumably because recognizing the attributes is harder in this dataset.

To gain further intuition, we compared the errors that *Causal* makes to those of *LE\**, the strongest no-prior baseline. With *Causal*, 39% of unseen pairs (U) are confused for seen pairs (S), and 36% of unseen pairs are confused for incorrect unseen-pairs. This yields a balanced rate of $\frac{U \to S}{U \to U} = \frac{39\%}{36\%} \approx$ 1.1. For comparison *LE\** errors are largely unbalanced: $\frac{U \to S}{U \to U} = \frac{67\%}{19\%} \approx 3.5$.

(a) Ground-truth = *(Suede, Slippers)*

(b) Ground-truth = *(Hair.Calf, Shoes.Heels)*

(c) Ground-truth = *(Patent.Leather , Shoes.Heels)*

Figure S.5: Qualitative examples of *Causal* success cases for Zappos: Every image was predicted correctly by *Causal*, but erroneously by *LE\**. Above every image we denote the erroneous prediction of *LE\**. (a) An unseen pair: the error rate was 40% for *Causal*, while it was 82% for *LE\**. (b) An unseen pair: the error rate was 39% for *Causal*, while it was 58% for *LE\**. (c) A **seen** pair: the error rate was 56% for *Causal*, while it was 83% for *LE\**.

We further analyzed wins and losses of *Causal* compared with *LE\**, which are illustrated in Figure S.5. *Causal* succeeded to overcome common failures of *LE\**, sometimes overcoming domain-shifts that exist within the seen pairs. The main weakness of *Causal* is that the error rate for seen pairs is higher compared to *LE\**. This effect is strongest for (1) $\big($A=*Leather*, O=*Boots.Ankle*$\big)$, which was mostly confused for either A=*Full.grain.leather* or A=*Suede*, while the object-class was correctly classified. (2) $\big($A=*Leather*, O=*Boots.Mid-Calf*$\big)$, which was mostly confused for A=*Faux.Leather* while the object-class was correctly classified. This result shows that *Causal* is less biased toward predicting *Leather*, which is the most common attribute in the training set.

## G.2 AO-CLEVr

We analyzed the errors that Causal makes when recognizing unseen pairs at the $5{:}5$ split[4] of AO-CLEVr (open). In $15\%$ of cases the object was predicted correctly, in $82\%$ the attribute was predicted correctly and in $3\%$ neither. It appears that in this case, recognizing the attribute transferred better to the unseen pairs, because in this dataset colors can be easily recognized.

Comparing the errors that Causal makes to those of *TMN\**, the strongest no-prior baseline show that with *Causal*, 19% of unseen pairs (U) are confused for seen pairs (S), and 20% of unseen-pairs are confused for incorrect unseen-pairs, a balanced rate of $\frac{U \to S}{U \to U} = \frac{19\%}{20\%} \approx 0.9$. However, with *TMN\**, confusion is largely unbalanced: $\frac{U \to S}{U \to U} = \frac{41\%}{13\%} \approx 3$.

- Select the attribute that **best** describes the **donut** under "Attribute 1". If none of the attributes fit, select the value "none of the above".
- Then, select the **second-best** attribute that fits the **donut** under "Attribute 2". If none of the attributes fit, select the value "none of the above".
- Note: "Attribute 1" should be a better fit than "Attribute 2".

**Select an option**

Attribute 1:
black

Attribute 2:
dented

Figure S.6: An example of a task on Amazon Mechanical Turk.

# H    Label-quality evaluation: Human-Rater Experiments

To better understand the level of label noise in *MIT-states* dataset we conducted an experiment with human raters.

Each rater was presented with an image of an object and was asked to select the best and second-best attributes that describe this object from a pre-defined list. The list was comprised of attributes that co-occur with the given object in the dataset. For example, the object "apple" had candidate attributes "green", "yellow" and "red", but not "black". Raters were also presented with an option "none of the above" and an option "I don't know", see Figure S.6. We sampled 500 instances of objects and attributes, one from each attribute-object pair. As a label-quality score, we computed the balanced accuracy of rater responses compared with the label provided by the dataset across the 500 tasks. To verify that raters were attentive we also introduced a "sanity" set of 30 instances of objects for which there were two clear attributes as answers. We also recorded the rate at which raters chose the "none of the above" and "I don't know" answers as a proxy for the difficulty of assigning labels to the dataset. Balanced accuracy was computed by averaging the accuracy per attribute.

The average rater top-1 and top-2 accuracies were 31.79% and 47% respectively, indicating a label noise level of ∼70%. The fraction of tasks that raters selected "none of the above" or "I don't know" was 5%, indicating that raters were confident in about ∼95% of their rating. The top-1 accuracy on the "sanity" set was 88% and the top-2 accuracy was 100%, indicating that the raters were attentive and capable of solving the task at hand.

Finally, Fig. S.7 shows qualitative examples for the label quality of MIT-States. For each label of 5 attribute labels, selected by random, we show 5 images, selected by random. Under each image, we show the choice of the amazon-turker in the label-quality experiment and the provided attribute label.

Figure S.7: Label quality of MIT-States. Showing 5 attribute labels, selected by random. For each label, we show 5 images, selected by random. For each image, we show the choice of the amazon-turker (AT) and the provided attribute label (Lbl). Green image margins indicate that the turker choice agrees with the label. Red margins indicate that the turker choice disagrees with the label.

# I  Numeric values for the metrics

## I.1  Overlapping split

| U:S | CAUSAL Unseen | Seen | Harmonic | Closed | VISPROD&CI Unseen | Seen | Harmonic | Closed | VISPROD Unseen | Seen | Harmonic | Closed |
|---|---|---|---|---|---|---|---|---|---|---|---|---|
| 2:8 | $77.7 \pm 1.4$ | $89.7 \pm 1.9$ | $83.2 \pm 1.2$ | $87.0 \pm 2.1$ | $60.0 \pm 2.7$ | $87.8 \pm 2.3$ | $71.1 \pm 1.7$ | $85.9 \pm 1.0$ | $42.5 \pm 3.0$ | $90.1 \pm 1.4$ | $57.5 \pm 2.9$ | $87.3 \pm 1.3$ |
| 3:7 | $72.2 \pm 1.0$ | $80.9 \pm 3.6$ | $75.7 \pm 2.3$ | $84.1 \pm 2.5$ | $44.7 \pm 5.0$ | $84.3 \pm 3.8$ | $58.1 \pm 4.9$ | $72.6 \pm 7.5$ | $29.2 \pm 3.7$ | $85.6 \pm 3.4$ | $43.2 \pm 4.4$ | $73.5 \pm 6.9$ |
| 4:6 | $67.4 \pm 2.0$ | $84.1 \pm 1.8$ | $74.7 \pm 1.7$ | $86.6 \pm 0.7$ | $53.2 \pm 1.5$ | $87.9 \pm 1.1$ | $66.2 \pm 0.9$ | $88.2 \pm 1.0$ | $35.9 \pm 3.6$ | $85.0 \pm 2.2$ | $49.8 \pm 2.9$ | $87.0 \pm 1.6$ |
| 5:5 | $47.1 \pm 4.5$ | $83.8 \pm 0.8$ | $59.8 \pm 3.9$ | $71.6 \pm 4.9$ | $38.3 \pm 1.1$ | $82.1 \pm 4.2$ | $52.1 \pm 1.8$ | $65.4 \pm 3.6$ | $18.9 \pm 0.2$ | $84.5 \pm 8.5$ | $30.3 \pm 0.8$ | $61.6 \pm 2.3$ |
| 6:4 | $26.9 \pm 0.5$ | $86.1 \pm 2.9$ | $40.9 \pm 1.0$ | $44.6 \pm 1.9$ | $20.0 \pm 1.8$ | $86.4 \pm 1.0$ | $32.3 \pm 2.4$ | $39.6 \pm 2.6$ | $11.1 \pm 1.4$ | $81.0 \pm 9.5$ | $18.8 \pm 1.5$ | $37.4 \pm 3.5$ |
| 7:3 | $22.8 \pm 3.0$ | $69.3 \pm 6.1$ | $33.7 \pm 4.2$ | $40.9 \pm 3.6$ | $15.7 \pm 2.5$ | $68.5 \pm 6.6$ | $25.1 \pm 3.4$ | $38.9 \pm 1.6$ | $14.1 \pm 1.5$ | $50.0 \pm 8.6$ | $21.6 \pm 2.4$ | $36.8 \pm 2.7$ |

| U:S | LE* Unseen | Seen | Harmonic | Closed | LE Unseen | Seen | Harmonic | Closed | ATTOP* Unseen | Seen | Harmonic | Closed |
|---|---|---|---|---|---|---|---|---|---|---|---|---|
| 2:8 | $72.1 \pm 2.8$ | $91.9 \pm 0.2$ | $80.7 \pm 1.7$ | $85.5 \pm 1.8$ | $72.7 \pm 3.3$ | $92.2 \pm 0.3$ | $81.2 \pm 2.0$ | $86.0 \pm 1.5$ | $78.4 \pm 3.3$ | $77.2 \pm 1.0$ | $77.6 \pm 1.4$ | $84.1 \pm 4.0$ |
| 3:7 | $41.9 \pm 6.3$ | $92.5 \pm 0.3$ | $56.8 \pm 6.4$ | $74.5 \pm 4.5$ | $42.1 \pm 6.9$ | $89.8 \pm 2.8$ | $56.6 \pm 7.2$ | $75.3 \pm 5.1$ | $62.5 \pm 2.3$ | $72.3 \pm 1.4$ | $66.9 \pm 1.2$ | $78.9 \pm 4.1$ |
| 4:6 | $57.8 \pm 4.5$ | $85.8 \pm 1.5$ | $68.6 \pm 2.7$ | $79.9 \pm 2.7$ | $47.2 \pm 5.7$ | $84.3 \pm 1.6$ | $59.7 \pm 4.5$ | $73.7 \pm 1.6$ | $68.8 \pm 1.2$ | $69.3 \pm 2.8$ | $68.8 \pm 0.9$ | $81.6 \pm 2.1$ |
| 5:5 | $31.7 \pm 1.6$ | $90.7 \pm 1.4$ | $46.8 \pm 1.9$ | $54.7 \pm 5.6$ | $26.3 \pm 2.1$ | $86.4 \pm 1.1$ | $40.2 \pm 2.6$ | $52.3 \pm 8.3$ | $44.3 \pm 1.1$ | $70.8 \pm 2.3$ | $54.4 \pm 1.2$ | $57.8 \pm 2.5$ |
| 6:4 | $20.1 \pm 1.0$ | $81.2 \pm 4.7$ | $31.9 \pm 0.9$ | $31.8 \pm 1.3$ | $20.8 \pm 2.4$ | $84.4 \pm 2.6$ | $33.1 \pm 2.8$ | $29.7 \pm 1.7$ | $25.8 \pm 1.9$ | $69.5 \pm 2.2$ | $37.4 \pm 1.7$ | $35.4 \pm 0.3$ |
| 7:3 | $13.7 \pm 2.2$ | $83.1 \pm 3.8$ | $23.1 \pm 3.3$ | $27.7 \pm 3.4$ | $11.4 \pm 3.2$ | $89.2 \pm 3.3$ | $19.7 \pm 5.1$ | $24.4 \pm 6.0$ | $10.6 \pm 1.7$ | $60.7 \pm 7.7$ | $17.9 \pm 2.7$ | $14.5 \pm 1.6$ |

| U:S | ATTOP Unseen | Seen | Harmonic | Closed | TMN* Unseen | Seen | Harmonic | Closed | TMN Unseen | Seen | Harmonic | Closed |
|---|---|---|---|---|---|---|---|---|---|---|---|---|
| 2:8 | $79.9 \pm 3.5$ | $77.9 \pm 1.1$ | $78.7 \pm 1.3$ | $84.2 \pm 3.9$ | $78.4 \pm 5.2$ | $87.2 \pm 0.8$ | $82.2 \pm 2.7$ | $87.9 \pm 1.5$ | $79.7 \pm 4.4$ | $85.8 \pm 0.9$ | $82.4 \pm 2.1$ | $88.7 \pm 1.6$ |
| 3:7 | $64.3 \pm 3.3$ | $72.7 \pm 1.8$ | $68.0 \pm 1.4$ | $77.0 \pm 3.9$ | $62.7 \pm 6.1$ | $86.5 \pm 0.4$ | $72.1 \pm 4.2$ | $81.7 \pm 5.0$ | $62.2 \pm 4.9$ | $86.5 \pm 0.3$ | $72.0 \pm 3.4$ | $81.4 \pm 5.0$ |
| 4:6 | $68.8 \pm 1.2$ | $68.7 \pm 3.1$ | $68.5 \pm 1.4$ | $83.0 \pm 2.0$ | $70.4 \pm 3.0$ | $83.0 \pm 3.2$ | $75.8 \pm 0.5$ | $86.6 \pm 1.3$ | $68.1 \pm 4.1$ | $83.8 \pm 2.6$ | $74.5 \pm 1.6$ | $86.5 \pm 2.5$ |
| 5:5 | $46.3 \pm 1.2$ | $67.7 \pm 2.5$ | $55.0 \pm 1.5$ | $59.9 \pm 2.8$ | $39.7 \pm 3.7$ | $84.9 \pm 3.7$ | $53.2 \pm 2.4$ | $67.8 \pm 5.1$ | $38.0 \pm 3.0$ | $84.7 \pm 2.2$ | $51.5 \pm 2.2$ | $67.3 \pm 4.7$ |
| 6:4 | $27.9 \pm 2.3$ | $72.4 \pm 2.7$ | $39.9 \pm 2.0$ | $40.2 \pm 0.3$ | $17.0 \pm 2.6$ | $87.9 \pm 1.2$ | $28.1 \pm 3.6$ | $40.0 \pm 0.9$ | $18.1 \pm 2.9$ | $83.7 \pm 0.4$ | $29.1 \pm 3.7$ | $41.7 \pm 0.7$ |
| 7:3 | $13.3 \pm 1.5$ | $56.0 \pm 5.2$ | $21.2 \pm 2.2$ | $19.2 \pm 2.1$ | $7.3 \pm 1.6$ | $93.1 \pm 3.3$ | $13.1 \pm 2.7$ | $35.4 \pm 2.2$ | $5.8 \pm 0.9$ | $88.1 \pm 2.5$ | $10.8 \pm 1.6$ | $36.7 \pm 1.0$ |

Table S.2: Numeic values for results of Figure S.2 (top row)

## I.2  Non-overlapping split

| U:S | CAUSAL Unseen | Seen | Harmonic | Closed | LE Unseen | Seen | Harmonic | Closed | ATTOP Unseen | Seen | Harmonic | Closed |
|---|---|---|---|---|---|---|---|---|---|---|---|---|
| 2:8 | $64.3 \pm 1.0$ | $79.4 \pm 1.5$ | $70.8 \pm 1.0$ | $82.1 \pm 0.7$ | $35.4 \pm 5.1$ | $80.1 \pm 3.7$ | $48.7 \pm 5.5$ | $71.2 \pm 2.6$ | $53.4 \pm 3.7$ | $67.8 \pm 3.2$ | $59.1 \pm 1.1$ | $76.0 \pm 5.4$ |
| 3:7 | $48.7 \pm 4.7$ | $75.3 \pm 4.6$ | $58.9 \pm 5.0$ | $79.0 \pm 5.5$ | $22.9 \pm 2.5$ | $84.5 \pm 2.3$ | $35.7 \pm 3.3$ | $52.5 \pm 6.8$ | $37.4 \pm 5.5$ | $65.6 \pm 2.4$ | $46.3 \pm 4.2$ | $61.1 \pm 10.9$ |
| 4:6 | $43.5 \pm 4.6$ | $69.2 \pm 4.2$ | $53.2 \pm 4.5$ | $66.8 \pm 5.3$ | $27.5 \pm 2.3$ | $80.7 \pm 2.9$ | $40.7 \pm 2.6$ | $43.4 \pm 4.2$ | $21.7 \pm 9.5$ | $49.2 \pm 5.3$ | $26.8 \pm 10.6$ | $40.9 \pm 10.1$ |
| 5:5 | $15.7 \pm 1.7$ | $75.2 \pm 7.3$ | $25.8 \pm 2.7$ | $37.5 \pm 4.9$ | $9.1 \pm 2.1$ | $91.1 \pm 0.5$ | $16.3 \pm 3.5$ | $19.9 \pm 3.6$ | $6.4 \pm 1.9$ | $65.3 \pm 7.5$ | $9.9 \pm 2.0$ | $22.0 \pm 4.2$ |

| U:S | TMN Unseen | Seen | Harmonic | Closed |
|---|---|---|---|---|
| 2:8 | $47.2 \pm 2.7$ | $82.1 \pm 2.9$ | $59.5 \pm 1.6$ | $81.2 \pm 2.2$ |
| 3:7 | $23.4 \pm 4.2$ | $84.2 \pm 0.2$ | $35.1 \pm 5.4$ | $64.4 \pm 5.6$ |
| 4:6 | $15.3 \pm 4.4$ | $85.5 \pm 3.1$ | $24.7 \pm 6.3$ | $54.4 \pm 4.7$ |
| 5:5 | $3.0 \pm 1.1$ | $86.8 \pm 3.4$ | $5.6 \pm 2.0$ | $32.7 \pm 1.6$ |

Table S.3: Numeric values for results of Figure S.1

## Footnotes

[2]When ($\hat{\phi}_{a'} \perp\!\!\!\perp O|A$), then by definition $p(\hat{\phi}_{a'}|a,o) = p(\hat{\phi}_{a'}|a)$. Therefore encouraging ($\hat{\phi}_{a'} \perp\!\!\!\perp O|A$), makes $p(\hat{\phi}_{a'}|a,o)$ approach $p(\hat{\phi}_{a'}|a)$

[3]We used 150 and not 100, in order to have total size of $2 \cdot 150 = 300$ for the concatenated representation of $[\hat{\phi}_a, \hat{\phi}_o]$. 300 is comparable to the default value for emb-dim in the LE baseline.

[4]The split used for ablation study