[Reviews · NeurIPS 2020]

Review 1

Summary and Contributions: ===== After Rebuttals Edit ========= After I went through all the reviews and author responses, I decided to raise my final rating from 6 to 7. However, R4 pointed out that the current method is most suitable for disentangled attributes. I think it could be the biggest limitation of this work, because most of the attributes manifest differently for different objects. Overall, I think this work is closely relevant to the NeurIPS community and their formulation is novel. ================================ In this paper, the authors provide a new formulation of compositional zero-shot recognition using a causal intervention. It considers the task as finding the best object-attribute embeddings that maximize the probability of the given image feature, i.e., argmax_{a,o} = p(x|a,o). The experiment results show that the proposed method can better recognize new unseen attribute-object compositions by better disentanglement.

Strengths: 1. By considering the compositional zero-shot recognition as generating image feature x from the independently sampled attribute and object embeddings, the proposed method achieves better disentanglement. Besides, the underlying attribute and object vocabulary embeddings are served as prototypes rather than features generated from the image. 2. The paper is well-written and easy to follow.

Weaknesses: 1. According to Eq.3, notations h_a, h_o, g(h_a, h_o) are the means of \phi_a, \phi_o and x, so my question is how do they update during training (they are also involved in the training losses)? Are they implemented as moving average during training or updated only after each epoch? 2. The disentanglement seems to be achieved by independence loss rather than intervention in the inference. Can I consider that the interventional inference means matching prototypes?

Correctness: Yes

Clarity: Yes

Relation to Prior Work: Yes

Reproducibility: Yes

Additional Feedback:


Review 2

Summary and Contributions: This paper proposed a new formulation for compositional zero-shot recognition by learning underlying disentangle structures of attribute compositions. To achieve this goal, the paper designed several reasonable regularization terms. They evaluated their approach on a synthesized dataset and a real world dataset and achieved SOTA performance compared to various baselines.

Strengths: The proposed approach is interesting and effective. From the causal perspectives, they model the process of how to generate an image by composing several attributes and elements. Figure 1 quite help to understand this point. Also, the designed regularization terms are reasonable and properly approximating the goals, where the latter points are well-illustrated in the appendix. The experimental description is very detailed and allows other researchers to replicate their studies. The proposed AO-CLEVr dataset can benefit the related research community, though it is a little bit too easy. Extensive experimental results also demonstrate the effectiveness of the proposed method.

Weaknesses: Results in the ablation studies indicate that the independence loss would encourage the performance on the unseen data but also significantly drop on the seen data. More discussion and analysis should be included here. Some failure cases would help understand the limitations of the proposed approach for the compositional generalization. The reviewer would appreciate if the author can discuss on some failure qualitative cases.

Correctness: Yes, the claims, method, and empirical methodology are correct.

Clarity: Yes, overall the paper is very easy to follow.

Relation to Prior Work: Yes, discussions in the related work section are sufficient.

Reproducibility: Yes

Additional Feedback: ---Post-rebuttal--- I thank author's response. Please include the mentioned discusses and qualitative results in the revision.


Review 3

Summary and Contributions: This work proposes a novel perspective for the compositional zero-shot learning, i.e, leveraging the causal operation intervention to disentangle the objects and attributes and further learning the generation of images as effects. The insight about the confounder within the attribute-object is interesting, and the experiments verify the efficacy of the proposed method well.

Strengths: + Causal view for the CZSL problem is non-trivial, and this paper reiterates the difficulty and importance of how to intervene and learn the intrinsic causal relationship from the raw data. + Using the HSIC to estimate the dependence between object and attribute is interesting. I quite like the analysis of the relation of L_indep and PIDA in the supplementary. + The constructed AO-CLEVr is a good attempt for the controlled compositional learning task. + Extensive details about the experiment are provided.

Weaknesses: - The method part needs a major revision to make the reading rhythm smoother. After the intro and related work, readers may be very curious about the detailed design and implementation of the intervention to disentangle the input attribute and object. But too much "in the supplementary" make the whole method illustration obstructed. And the whole method sec is not depicted well and lacks clarity, especially the most important "independent" part. - The claim about the MIT dataset is still open for me. Because MIT is under the single attribute set for each object, and each object usually can have many attributes. Hence, let the annotators choose a single attribute maybe not very reasonable. Even the annotations are different from the labels, this may be caused by the ambiguity of the attribute-object itself. At least the top-k matching ratio should be considered. Furthermore, in the supplementary, the samples of the annotations cannot prove the claim very strongly for me. For example, ancient/burnt castle in the 2nd row, modern/empty room in the 4th row, ripe/sliced apple and raw/sliced meat, and cooked/sliced food in the last row are all reasonable in my opinion. These cases can well depict the ambiguity of the attribute comes from the language. Thus, I think it is still essential to give a comparison between this work and previous sota methods in MIT states. Because previous methods all show obvious similar performance trends on MIT-states and UT-Zappos. - The attributes which do not have obvious physical meanings like cute, comfortable (COCO-A) maybe not matched with the assumption of this work (L149). And this work is hard to be compatible with the multiple attributes and single object setting. - The assumption of Gaussian is too simplified for me. But the authors also honestly discuss this.

Correctness: Basically yes. The claim about the MIT needs more discussion in my opinion.

Clarity: Not very well.

Relation to Prior Work: Yes.

Reproducibility: Yes

Additional Feedback: 1. If using the mutual information instead of the HSIC, how will the performance and training change? 2. Does the capacity or embedding size affect the intervention? Please give some discussion about the efficacy of HSIC in training. 3. Why the performances with and without the alternate training so different? 4. The whole paper needs a major revision to keep the reading smoother. Some content should be reorganized, e.g., many important discussions are better to be put in the main paper. And the insight part of the main paper can be more concise to leave some space for the above details. ---Post-rebuttal--- After reading all the reviews and the author's response, some of my concerns are addressed. Although I am still a little bit concern about the claim about the MIT dataset, I tend to maintain my score. And I agree with the issues from R4, especially about the clarity of the method and equations. I think a major revision of the method part is very essential to clarify the method details, eg., intervention implementations, loss analysis, the experiment discussion, especially the limitation about the assumption and the attribute setting.


Review 4

Summary and Contributions: This paper proposes an attribute-object composition learning method that is inspired by a causal inference based interpretation of the data generation/composition process. The interpretation is that attributes and objects generate prototype distributions that are independent of each other, and they in turn together generate images of compositions. This translates to a simple, fixed DAG where attribute and object features have single parents (the attribute class, and object class respectively), and are both parents to the image features. This structure admits a particular set of compositions (simple colors/textures + objects) and the structure is encoded as losses that try to enforce independence constraints (among other regularizers) and the authors apply this to a standard embedding learning framework. The proposed model outperforms prior methods on two datasets, and across a wide variety of metrics.

Strengths: * The premise behind assigning a causal interpretation to the data generation process of compositions is very interesting. The kinds of compositions that can be targeted with the specific structure proposed in the paper may be limiting, but the general direction of this work is strong. * The proposed model achieves strong results on two datasets. The authors do a very thorough job accounting for all metrics previous work typically relies on, including recent evaluation methods proposed in [40]. The ablations in Supp are also thorough.

Weaknesses: * This method is most suitable for variables that have a single parent in the causal DAG -- the class label. This severely restricts the class of attributes that can be modeled and manifests in the paper as experiments with simple attributes (colors in AO-CLEVr, and materials in Zappos). In fact, prior work has noted that attributes (or other compositional modifiers) manifest very differently for different objects ([36] gives the examples from prior work: "fluffy" for towels vs. dogs, "ripe" for one fruit vs. another etc.). For these attributes, and many others, the data generating process is not so straightforward -- there are edges from both attribute labels and object labels to the core features. The authors do acknowledge this limitation in L326, however it is an important weakness to consider given that _difficult_ instances in real world datasets (where both object and attribute are parents of \phi_a for example) are fairly prevalent. * I'm a little unclear on some details surrounding the mapping from the causal graph to the mappings in Fig (c). A few clarifications: - I do not fully understand the role of the functions g_A^{-1} and g_O^{-1} in the causal graph. The SEM equations in Supp A.1 fully represent the relationships between the variables presented and does not seem to accommodate the extra two functions. Does the inclusion of these functions make additional assumptions on the structure? What do they mean in the context of the existing DAG? - Training these inverse models also seems to go against L130: "... that treats labels as causes, rather than as effects of the image" which is the standard embedding learning approach. I think the point where the analogy starts to break down is the assumption that the prototype feature can be approximated with the inferred feature from the image. This is equivalent to creating two new nodes in the graph (\hat{\phi_a}) and (\hat{\phi_o}), who each have a single parent x. These aren't the same as the nodes \phi_a, and \phi_o (as these are unobserved), and since their parents are only x, they already by construction satisfy the independence constraints. It would be helpful if more detail is provided for why this is consistent with the original proposal. - Given that phi_a, phi_o and x are all represented as multivariate gaussians, for the SEM model with independent noise, do any assumptions fail to hold when the functions relating these distributions are implemented as nonlinear MLPs? * In the final implementation, it is difficult to see where the causal interpretation developed through the paper actually manifests. It would help if the rebuttal contained a clear-cut difference between a standard embedding learning approach and the proposed method. As far as I can tell, the main difference is the inclusion of the independence loss which is inspired from the structure of independences encoded in the proposed DAG. All the other loss functions are commonly explored in prior work: the data likelihood loss (L184) is identical to ones used for embedding learning methods, and the invertible embedding loss (L208) is the same as an auxiliary classification loss in prior work. Is this interpretation correct? - Another place where this confusion arises is in the ablation experiments in Supp Table S2 where weights of various losses are set to zero and models are retrained. Here, it appears that \lambda_{invert} is an essential component of the model, and removing it produces results that are far worse than removing the independence loss. What could be the reason for this? Moreover, it seems like this is a regularizer that has no relation to the original causal interpretation proposed.

Correctness: -

Clarity: * The paper is written well. It is easy to follow for the most part. The method overview in Sec 3. helps put standard embedding learning models in perspective with the interpretation being presented here. * Some bits are confusing and could do with a little bit more explanation: - The HSIC loss is an important component of the independence constraints, but its actual form has not been specified in this paper. For clarity, it could be a good idea to explicitly express this equation in terms of the variables presented for this application if space permits (or at least in Supp). Reproduciblity may be an issue if this is not described in detail. - Fig 1a. What are some hypotheses on what these confounders could be? I understand these are unobserved, but are there any possible theories about what they are given the setup of attribute-object composition?

Relation to Prior Work: * The paper does not discuss much work related to causal interpretations for other applications in computer vision beyond domain adaptation. For example: - CausalGAN (Kocaoglu et al. ICLR18) - Causal Confusion in Imitation Learning (de Haan et al. NeurIPS19) - Discovering Causal Signals in Images (Lopez-Paz et al. CVPR17)

Reproducibility: Yes

Additional Feedback: Questions: * While it is beyond the scope of this paper, if there is space in the rebuttal, I would like to hear the authors thoughts on causal models for more complicated structures over attributes/objects (or structures that have to be inferred from data). This is referring to the comment in L148-150 "For attributes which have no physical meaning in the world, it may not be possible...". What could be done for these cases? Overall, it is unfortunate that the stipulated DAG is fixed and only admits certain attributes (colors, textures) that on the surface may seem trivial to identify from images, but the strong results on the two datasets compared to prior work on composition learning does merit consideration. I am leaning towards positive modulo some concerns outlined in the weakness section. ------------------------------------- Updated review: ------------------------------------- Thanks to the authors for the rebuttal. I have read it and gone through the other reviews. I appreciate the insights about future directions the authors provide, and the clarifications about the contributions compared to the standard embedding learning setup. I am still a bit unclear about how estimating the core features from the images using the inverse functions works in the context of the causal structure, but the other reviewers have not raised any objection. I am also not convinced about how this method can extend to a more relevant set of natural images (with different causal structures), but again, the authors have been honest about this limitation in both the paper and the rebuttal. I maintain my original rating as the results are strong on the proposed task.

[Author Response · NeurIPS 2020]

We thank the reviewers for their insightful and valuable feedback and for their *unanimous support* of the paper.
We are encouraged that they found our formulation to be **"novel/new"** (**R1**,**R2**,**R3**) **"interesting"** (**R2**,**R3**,**R4**) and
with a **"strong direction"** (**R4**); The experimental results to be **"effective"** (**R1**,**R2**,**R3**,**R4**), **"extensive"**,**"thorough"**
(**R2**,**R4**), **"detailed"** (**R2**,**R3**) and **"strong"** (**R4**); And AO-CLEVr dataset **"beneficial"**,**"good"** (**R2**,**R3**).

**R4: DAG is most suitable to disentangled attributes, some attribute manifest differently depending on object.** Extend-
ing to dependent pairs is a very important next direction. Unfortunately, it is not clear that zero-shot can work well
with strongly entangled pairs because every case could be special. Three insights worth mentioning: (1) Even the
fully disentangled case is still very challenging. (2) The factored DAG can be used as a strong zero-shot prior for
few-shot learning, thus benefiting future work on dependent pairs. (3) Using a "closed" settings may capture some of
the dependency by eliminating "over-generalization" (e.g. by disallowing yellow-wine label).

**R4: Clarify details of mapping the causal graph to Fig 1c (1) The role of $g_A^{-1}, g_O^{-1}$, do they add assumptions? context**
**to causal DAG?** $g_A^{-1}$ and $g_O^{-1}$ are used to estimate the latent $\phi_a$ and $\phi_o$ of an image instance. They reflect as-
sumptions about the noise level in the data-generation process (Suppl L508-513), i.e. that the mapping from the
core-features ($\phi_a$ and $\phi_o$) to the image (**x**) is not too noisy and the latent vector can be recovered from the image.
**(2) The new nodes $\hat{\phi}_a, \hat{\phi}_o$ satisfy the independence constraints by construction. Explain consistency.** We respectfully
point out that since $\hat{\phi}_a, \hat{\phi}_o$ are children of **x**, they *do not* satisfy the independence constraints of Eq. 6. Minimiz-
ing $L_{indep}$ encourages the property $p(\hat{\phi}_o|do(o)) \approx p(\hat{\phi}_o|do(a,o))$ (L550). Only then the independence relations of Eq.
(6) apply to $\hat{\phi}_a, \hat{\phi}_o$. It also minimizes the PIDA metric of (Suter 2019). **(3) Any assumptions fail for MLPs?** No.

**R4: Where does the causal interpretation manifests? (1) Difference from standard embedding?** As the reviewer points
out, one difference is in the independence loss; another is the use of two separate embedding terms tied to the in-
dependence loss; both motivated by the causal graph. We deliberately proposed a model close to baselines (L186)
to surgically demonstrate the strength of the proposed approach. **(2) Why is $\lambda_{invert}$ essential?** Since there exist no
ground truth values for neither $\phi_a$ nor $h_a$, minimizing $||\hat{\phi}_a - h_a||^2$ may reach trivial solutions (same for $\phi_o, h_o$).
$\lambda_{invert}$ guides the optimization and avoids trivial solutions. It does not contradicts assumptions on the causal process.

**R1: How are the means $h_a, h_o, g(h_a, h_o)$ updated?** Instead of learning explicit values for the means, we learn MLPs
that output the means (using gradient updates L138,143,178). For example, an MLP ($h_A$) maps the (one-
hot) representation of "leather" to $h_{leather}$ and an MLP ($g$), maps ($h_{leather}, h_{sandal}$) to $g(h_{leather}, h_{sandal})$.
**R1: Does interventional inference means matching prototypes?** Partially yes: Inference that follows *the approxima-*
*tions we took* (Supp A., e.g. Gaussian and $0^{th}$ order Taylor) may be viewed as matching prototypes. In the general
case, there may be better ways to estimate the likelihood of $p(\mathbf{x}|a,o)$ and the factors $p(\phi_a|a), p(\phi_o|o), p(x|g(\phi_a,\phi_o))$.
**R1: Disentanglement is achieved by independence loss rather than intervention.** The independence loss allows to
learn a model that is robust to interventions. Minimizing $L_{indep}$ encourages $p(\phi_o|do(o)) \approx p(\phi_o|do(a,o))$ (L550).

**R2: Independence loss encourages the performance on the unseen data but drops on the seen data.** This is a known
and important trade-off (Rothenhäusler 2018): The independence loss discourages certain types of correlations, hence
models do not benefit of them when the test and train distributions are identical. However, the loss is constructed
in such a way that these are exactly the correlations that fail to hold once the test distribution changes (zero-shot).
Ignoring these correlations improves performance on unseen data. We will refer to (Rothenhäusler 2018) and discuss.
**R2: Failure analysis.** Following this request, we analyzed samples of unseen pairs of Zappos in the open-world setup.
We compared *Causal* with LE*, which is the strongest no-prior baseline. LE* confuses unseen pairs for seen pairs
at a rate of $3.7:1$, while *Causal* errors are more balanced $1.2:1$. One interesting failure case of *Causal*, is that it
over-commits for predicting the pair "Leather-Slippers", which was unseen during training. In the final version we
will provide more qualitative and quantitative details about Zappos and AO-CLEVr .

**R3:MIT dataset: Consider top-k labels**: Following this suggestion, we conducted a new experiment to evaluate both
top-1 and top-2 accuracy. Raters were asked to select the best and 2nd-best attributes that describe an image, among
attributes relevant for that object. The top-1 accuracy was 32%, consistent with previous experiment. The top-2
accuracy was 47%, only slightly higher than adding a random label on top of top-1 label (yielding 42%). To verify
that raters were attentive, we also injected 30 "sanity" questions that had two "easy" attributes, yielding top-2=100%.
**R3: MI instead of HSIC:** HSIC advantage is that it is non-parametric, unlike MI, and does not requires training
an additional network for variational approximation. **Embed. size**; We will report results w.r.t. embedding size.
**Efficacy of HSIC:** See $\lambda_{indep}$=0 at Table S.2. **Results w/o alternate training.** Alternate training lowers the SEM (L713).
Means are comparable (68.7 vs 67.7).

**R3: Revise method for smoother reading. R4 Some bits are confusing.** We will restructure the paper based on your
feedback: (1) Shorten the "overview" section (2) Discuss how independence loss allows $\hat{\phi}_a, \hat{\phi}_o$ to recover the proper-
ties of $\phi_a, \phi_o$, and its relation to PIDA. (3) Update the final version based on the rebuttal.

**R1**,**R2**,**R3**,**R4**: We will address all minor comments, and clarify the broader impact.

[Meta-Review · NeurIPS 2020]

All four reviewers appreciated the neat idea contained in this paper which is also shown to work well in practice. The authors open up the way for studying data generation processes through causal interventions, which is a novel and technically interesting direction. Most importantly, it is a significant direction which is expected to stimulate further research in the field. I am recommending acceptance of this paper, however please consider revising the manuscript to address R4’s remarks about clarity and R2’s and R3’s remarks about deeper discussion of failure cases and limitations.